



# Dramatic increase of reactive VOC emission from ships at berth after implementing the fuel switch policy in the Pearl River Delta Emission Control Area

Zhenfeng Wu[1,3], Yanli Zhang[1,2,*], Junjie He[4], Hongzhan Chen[4], Xueliang Huang[1,5], Yujun Wang[4], Xu Yu[1,3], Weiqiang Yang[1,3], Runqi Zhang[1,3], Ming Zhu[1,3], Sheng Li[1,3], Hua Fang[1,3], Zhou Zhang[6], Xinming Wang[1,2,3]

[1]State Key Laboratory of Organic Geochemistry and Guangdong Key Laboratory of Environmental Protection and Resources Utilization, Guangzhou Institute of Geochemistry, Chinese Academy of Sciences, Guangzhou 510640, China
[2]Center for Excellence in Regional Atmospheric Environment, Institute of Urban Environment, Chinese Academy of Sciences, Xiamen 361021, China
[3]University of Chinese Academy of Sciences, Beijing 100049, China
[4]Guangzhou Environmental Monitoring Center, Guangzhou 510640, China
[5]Yunfu Total Pollutant Discharge Control Center, Yunfu 527300, China
[6]Changsha Center for Mineral Resources Exploration, Guangzhou Institute of Geochemistry, Chinese Academy of Sciences, Changsha 410013, China

*Correspondence to*: Yanli Zhang (zhang_yl86@gig.ac.cn)





**Abstract.** Limiting the fuel sulfur content (FSC) is a widely adopted approach to reduce ship emissions of sulfur dioxide ($SO_2$)
and particulate matters (PM) particularly in emission control areas (ECA), but its impact on the emission of volatile organic
compounds (VOCs) is still not well understood. In this study, emissions from ships at berth in Guangzhou, south China, were
characterized before and after implementing the fuel switch policy with a FSC limit of 0.5% in the Pearl River Delta ECA in
south China. After implementing the fuel switch policy, the emission factors (EFs) of $SO_2$ and $PM_{2.5}$ for coastal vessels dropped
by 78% and 56% on average, respectively; the EFs of non-methane hydrocarbons (NMHCs), however, reached 1807 ±1746
mg/kg, about 15 times that of 118 ±56.1 mg/kg before implementing the new policy. This dramatic increase in the emission
of NMHCs might be largely due to the replacement of high-sulfur residual fuel oil with low-sulfur diesel or heavy oils, which
are typically more rich in short-chain hydrocarbons. Moreover, reactive alkenes overtook alkanes to become the dominant
group among NMHCs and low carbon number NMHCs, such as ethylene, propene and isobutane, became the dominant species
after the new policy. As a result of the largely elevated EFs of reactive alkenes and aromatics after the new policy, for per
kilogram of fuel burned, emitted NMHCs had nearly 29 times larger ozone formation potentials (OFPs) and about 2 times
higher secondary organic aerosol formation potentials (SOAFPs). Unlike coastal vessels, river vessels in the region used diesel
fuels all along and were not affected by the fuel switch policy, but their EFs of NMHCs were even 90% larger than that of
coastal vessels after implementing the new policy, with about 120% larger fuel-based OFPs and 70-140% larger SOAFPs. The
results from this study suggest that while the fuel switch policy could effectively reduce $SO_2$ and PM emissions and thus help
combat PM2.5 pollution, it would also lead to greater emissions of reactive VOCs, that may threatens ozone pollution control
in the harbor cities. This change for coastal or ocean-going vessels, along with the large amounts of reactive VOCs from river
vessels, raises regulatory concerns for ship emissions of reactive VOCs.
**1 Introduction**
World seaborne trade volumes are estimated to have accounted for over 80% of total world merchandise trade (UNCTAD,
2016). The controls on ship emissions, are however far less stringent than on land emission sources, and it is no surprise that
ship engines are among the world's highest polluting combustion sources in terms of per ton of fuel consumed (Corbett and
Fischbeck, 1997). As a large amount of marine ship emissions occur within 400 km of coastlines (Fu et al., 2017), ship
emissions would give rise to air pollution in coastal areas, and thus contribute substantially to environmental burden of disease
(Corbett et al., 2007; Lv et al., 2018; Feng et al., 2019; Ramacher et al., 2019; Wang et al., 2019a). Therefore, global efforts
have been made to regulate and prevent health risks from ship emissions particularly in harbor cities.
An important intervention policy by the International Maritime Organization (IMO) for reducing ship emissions is the
designation of emission control areas (ECA) where more stringent limit of fuel sulfur content (FSC) is implemented (IMO,
2017). This ECA approach has brought about significant improvements in ambient air quality for coastal areas (Lack et al.,
2011; Tao et al., 2013; Contini et al., 2015; Zetterdahl et al., 2016). In the North Sea regions, for example, the new policy
restricting FSC below 1.5% since 2007 resulted in reduction rates of 42%, 38% and 20%, respectively, for ambient



concentrations of sulfur dioxide ($SO_2$), sulphate aerosols and ammonium aerosols that were related to ship emissions (Matthias
et al., 2010); monitoring in U.S. coastal states revealed significant reductions in ambient $PM_{2.5}$ (particulate matter with an
aerodynamic diameter less than 2.5 μm) from residual fuel oil (RFO) combustion due to marine vessel fuel sulfur regulations
in the North American Emissions Control Area (NA-ECA) (Kotchenruther, 2017); in the Marmara Sea and the Turkish Straits,
ship emission of $SO_2$, $PM_{2.5}$ and $PM_{10}$ (particulate matter with an aerodynamic diameter less than 10 μm) were projected to
reduce by 95%, 67% and 67%, respectively, if FSC were restricted below 0.1% (Viana et al., 2015). Consequently, with the
increasingly stringent control over land-based emission sources, limiting ship emissions has gradually stood out as an effective
measure to combat air pollution in coastal zones.
Intervention measures on ship emission, however, are mostly targeted on $SO_2$ and PM, and much less attention has been
paid to other pollutants from ship emissions, such as nitrogen oxides ($NO_x$) and volatile organic compounds (VOCs), although
they are also important precursors to ozone and secondary aerosols (Chameides et al., 1992; Odum et al., 1997; Atkinson,
2000; O'Dowd et al., 2002). Cooper et al. (1996) found that many reactive VOCs, like ethylene, propylene and isobutylene,
were present in emission from passenger ferries in the Skagerak-Kattegatt-öresund region; Agrawal et al. (2008) reported
emissions of VOCs including carbonyls, 1, 3-butadiene, aromatics and n-alkanes from the main engine, auxiliary engine and
boiler of a Suezmax class vessel; Agrawal et al. (2010) and Murphy et al. (2009) further calculated their emission factors based
on shipboard platform measurements and aircraft-based measurements for the main engine of a PanaMax Class container ship.
Very recently, Huang et al. (2018a) tested a Handysize-class bulk carrier under at-berth, maneuvering and cruising condition,
and found single-ring aromatics accounted for 50-74% of VOCs with toluene as the most abundant species; Xiao et al. (2018)
tested 20 ships at berth in the Jingtang Port in north China and found that alkanes and aromatics instead dominated in the
VOCs emissions. As a matter of fact, previous studies have already demonstrated that ship emissions were able to impact the
ambient ozone formation in coastal cities (Wang et al., 2019b). Meanwhile, ship emissions could contribute substantially to
$NO_x$ in the oceans and coastal areas (Song et al., 2010; Tagaris et al., 2017). So, even for lowering ambient ozone levels, there
is a growing concern about ship emissions of ozone precursors, including $NO_x$ and VOCs.
China hosts many of the world's busiest ports, sharing about 10% of global ship emissions (Fu et al., 2017). To reduce ship
emissions, China has also designated three ECAs, namely the Pearl River Delta (PRD), the Yangtze River Delta and the Bohai
Rim, where ships are required to gradually switch to fuels with a FSC limit of 0.5% from January 1, 2017 to December 31,
2019. As estmiated by Liu et al. (2018), this fuel switch policy could lower atmosheric concentrations of $SO_2$ and $PM_{2.5}$ by
9.5% and 2.7%, respectively, in the coastal region of the PRD in south China. A recent field observation campaign in Jingtang
port also reveal that, due to the implementation of the fuel switch policy, ambient levels of $SO_2$ dropped from 165.5 ppb to
67.4 ppb while particulate vanadium (V), a marker of ship PM emission (Agrawal et al., 2009; Pey et al., 2013; Perez et al.,
2016; Tao et al., 2017), decreased drastically from 309.9 ng/m$^3$ to 9.1ng/m$^3$ (Zhang et al., 2019). However, it is still unknown
whether the fuel switch policy will bring about changes in ship emissions of VOCs.
For ships at berth, their main engines are shut down and auxiliary engines become the only emission source. As a ship is
usually at berth for one day or more and the place where its auxiliary engine discharges pollutants is usually closer to densely



populated areas, so emissions from ship at berth could have a larger impact on coastal areas (Cooper et al., 2003). In the present
study, we conducted shipboard platform measurements of air pollutants emitted from coastal vessels at berth in Guangzhou
Port in the PRD region in south China in 2017 and 2018 after implementing the fuel switch policy, and compared the results
with those from a similar campaign previously conducted also by us in 2015 and 2016 before implementing the fuel switch
policy. Apart from emissions of pollutants like $PM_{2.5}$ and $SO_2$, in this study we will put our focus on emissions of VOCs and
aim to investigate changes in composition profiles and emission factors of VOCs from ships at berth and to assess the potential
influence on the formation of ozone ($O_3$) and secondary organic aerosol (SOA) due to the fuel switch policy. Besides, river
vessels, which commonly use diesel oil as fuel and did not need to implement the fuel switch policy at the moment, were also
tested in 2017 in comparison with the coastal vessels that had implemented the policy.
**2 Experimental section**
**2.1 Study area**
Guangzhou Port is located in the estuary of the Pearl River and the center of the PRD region, adjacent to Hong Kong and
Macao (Fig. 1). In 2017, cargo throughput of Guangzhou Port reached 590 million tons, ranking the fifth in China and the
sixth in the world; and the container throughput in Guangzhou port reached 20.37 million TEU, ranking the fifth in China and
the seventh in the world (China Port Press, 2018). In 2013, Guangzhou Port was estimated to contribute near 40% ship
emissions of $SO_2$, $NO_x$, CO, $PM_{10}$, $PM_{2.5}$ and VOC from nine port groups in the PRD bay area (Li et al., 2016a).
**2.2 Test ships and fuel types**
As required, FSC for ships at berth should be lower than 0.5% since 1 January 2017. In the PRD, measures are even more
stringent that ships at berth should use diesel oil that conforms to Chinese national standard GB252-2015 (Standards Press of
China, 2015). Table 1 presents the basic information of 11 tested ships, among which ships C and D were tested both before
and after the implementation of the fuel switch policy. According to the classification of ships as by Li et al. (2016a), ships H,
I, J and K were river vessels, which were not regulated because they already used diesel oil as fuel before implementing the
fuel switch policy, and others were coastal vessels and none ocean-going ships were tested in this study.
**2.3 Portable emission measurement system**
The ship exhaust sampling system is composed of flue gas dilution system, flue gas analyzer, particulate matter sampler and
air sampler (Figure 2). The ship exhaust first enters the Dekati® ejector dilutor (DI-1000, Dekati Ltd., Finland) from the
sampling nozzle, spilt into four ways after diluted with clean air: one for air sampling by canisters and Teflon bags after passing
through a filter; the other two for collecting $PM_{2.5}$ samples with 47mm Teflon filters (Whateman, Mainstone, UK) and 47mm
quartz fiber filters (Whateman, Mainstone, UK), respectively, after the diluted exhaust was mixed well in a stay cabin, and
then passing through a $PM_{2.5}$ cutting head; the last is the vent. Before dilution, trace gases in the ship exhaust was directly



measured by a flue gas analyzer (F-550, WOHLER, Germany), and air sample was also collected simultaneously by canisters
and Teflon bags. The dilution ratios of the flue gas dilution system were then can be more accurately calculated by comparing
$CO_2$ concentrations in samples before and after the dilution. In addition, 500ml fuel oil being used by each ship was collected
in brown glass bottles for determining its carbon and sulfur contents, and analyzing $C_{11}$-$C_{36}$ hydrocarbon species.
Non-methane hydrocarbons (NMHCs) in air samples collected in canisters and Teflon bags were analyzed by using a Model
7100 Preconcentrator (Entech Instruments Inc., California, USA) coupled to an Agilent 5973N gas chromatography-mass
selective detector/flame ionization detector (GC-MSD/FID, Agilent Technologies, USA). More details about the analysis were
described elsewhere (Zhang et al., 2013; 2015); $CO_2$/CO concentrations were analyzed by gas chromatography (Agilent
6980GC, USA) with a flame ionization detector and a packed column (5A molecular sieve, 60/80 mesh, 3 m ×1/8 in.) (Liu et
al., 2015). The particulate samples collected by quartz filters were analyzed by a DRI Model 2015 multi-wavelength
thermal/elemental carbon (OC/EC) analyzer (Li et al., 2018). The carbon contents of ship fuels were analyzed by an elemental
analyzer (Vario EL III, Elementar, Germany), and the sulfur contents were analyzed by the conversion to sulfate with an
Oxygen Bomb Combustion (IKA AOD1, IKA, Germany) followed by determination of sulfate with an Ion Chromatography
(883 Basic IC plus, Metrohm, Switzerland) (Li et al., 2016b). $C_{11}$-$C_{36}$ hydrocarbons in fuels were analyzed with an Agilent
7890/5975C gas chromatography/mass spectrometer detector (GC/MSD) equipped with a HP-5MS capillary column (30 m in
length, 0.25 mm I.D., 0.25 μm film thickness) (Yu et al., 2018) after dissolving 50μl fuel oil in 1ml n-hexane and removing
the insoluble compositions through filtration.
**2.4 Calculations of emission factors**
Emission factors (EFs) were calculated by carbon balance approach, which assumes that the carbon in fuel is transformed into
the carbon in $CO_2$, CO, PM and VOCs, the EF of $CO_2$ is determined as following (Liu et al., 2014):
$$EF_{CO2} = \frac{C_F \cdot \Delta[CO_2]}{\Delta C_{CO2} + \Delta C_{CO} + \Delta C_{PM} + \Delta C_{VOCs}} , \qquad (1)$$
where $EF_{CO2}$ is the emission factor of $CO_2$ in grams per kilogram of fuel burned (g kg$^{-1}$); $C_F$ is the mass fraction of carbon in
fuel (g kg$^{-1}$); $\Delta[CO_2]$ is the incremental concentrations of $CO_2$; $\Delta C_{CO2}$, $\Delta C_{CO}$, $\Delta C_{PM}$, $\Delta C_{VOCs}$ represent the carbon mass
concentrations of $CO_2$, CO, PM and VOCs after subtracting their background concentrations.
The EF of a pollutant $i$ is calculated by:
$$EF_i = \frac{\Delta[i]}{\Delta[CO_2]} \times EF_{CO2} , \qquad (2)$$
where $\Delta[i]$ is the incremental concentrations of pollutant $i$.
According to the standard method ISO 8178-1, the sulfur in fuel is assuming to be fully transformed into $SO_2$, so we use Eq.
(3) to calculate the EF of $SO_2$ (Zhang et al., 2018a):
$$EF_{SO2} = S\% \times \frac{64}{32} \times 10^3 , \qquad (3)$$
where $EF_{SO2}$ is the EF of $SO_2$ in g kg$^{-1}$, and S% represents FSC.



**3 Results and discussion**

**3.1 Changes in EFs for ships at berth**

The FSC for the tested coastal vessels on average decreased from 2.2 ±0.5% before to 0.4 ±0.5% after implementing the fuel switch policy, though there are some ships, like ship G, violating the regulation with FSC still above the limit of 0.5% (Table 1). As a matter of fact, the ship fuel had transferred from residual fuel oil to diesel oil or heavy oil (Fig S1), and the compositions of fuels used by the coastal vessels tended to have more low-carbon number hydrocarbons as demonstrated by their total ion chromatograms (Fig S2). This change in fuel compositions may also explain why the mass percentages of <C6 VOCs (VOCs with carbon numbers below 6) in total VOCs in ship exhaust increased from 8.5%-27.3% to 44.4%-86.6% after implementing the fuel switch policy(Fig S3).

As shown in Table 2, the EFs for $SO_2$, which are independent of the combustion system (Corbett et al., 1999), decreased by 78.0% from 44.0 ±10.5 g kg$^{-1}$ to 9.66 ±7.97 g kg$^{-1}$ on average. Fuel-based EFs for $CO_2$, CO, $NO_x$, NMHCs, $PM_{2.5}$, OC and EC, however, are much more complicated as they are not only related to properties of the fuels, but also heavily influenced by performance of combustion system. The comparison before and after implementing the fuel switch policy is also challenged by the fact that the tested coastal vessels during the two campaigns are not the same ones and that we have tested a very limit number of ships. Nevertheless, ships C and D had been tested both before and after the new policy and we can make a comparison for them. The EF of $CO_2$ for ships C and D slightly increased from 3025 g kg$^{-1}$ and 3069 g kg$^{-1}$ before to 3131 g kg$^{-1}$ and 3196 g kg$^{-1}$ after the new policy; the EF of CO for ship C increased from 3.80 g kg$^{-1}$ to 6.16 g kg$^{-1}$, but that for ship D decreased from 14.6 g kg$^{-1}$ to 6.41 g kg$^{-1}$; the EF of NOx for ship C slightly decreased from 19.9 g kg$^{-1}$ to 19.0 g kg$^{-1}$, while that for ship D decreased from 51.5 g kg$^{-1}$ to 31.1 g kg$^{-1}$.

Like EFs of $SO_2$, the EFs of $PM_{2.5}$ also decreased significantly after the new policy. For example, EFs of $PM_{2.5}$ for ship C decreased by 45.6% from 1.02 g kg$^{-1}$ to 0.56 g kg$^{-1}$ and that for ship D decreased by 64.5% from 2.44 g kg$^{-1}$ to 0.87 g kg$^{-1}$; similar to that of $PM_{2.5}$, the EFs of OC and EC for ship C decreased by 28.7% and 56.1%, and that for ship D decreased by 60.5% and 63.0%, respectively. Therefore, after implementing the new policy, the changes in EFs of $CO_2$, CO and $NO_x$ were not significant for coastal vessels, but the EFs of $SO_2$, $PM_{2.5}$ and carbonaceous aerosols did become lower.

Compared to $SO_2$ or other pollutants, NMHCs from coastal vessels showed more dramatic changes in their EFs. As showed in Table 2, EFs of NMHCs ranged 60.7-197mg kg$^{-1}$ with an average of 118 ±56.1 mg kg$^{-1}$ before, and they ranged 292-5251mg kg$^{-1}$ with an average of 1807 ±1746 mg kg$^{-1}$ after implementing the fuel switch policy. For ships C and D that were tested both before and after the new policy, the EF of NMHCs for ship C increased about 6 times from 106 mg kg$^{-1}$ to 706 mg kg$^{-1}$, and that for ship D also increased about 4 times from 60.7 mg kg$^{-1}$ to 292 mg kg$^{-1}$. This great change in our study was consistent with that based on shipboard platform measurements by Copper et al. (2003), who also found the EFs of hydrocarbons from passenger ferry at berth increased from 0.29-0.57 g kg$^{-1}$ to 1.71 g kg$^{-1}$ after replacing the residual oil (FSC=0.53%) marine gasoil (FSC=0.09%) (Table 3).


There are only a few previous studies available about air pollutants from coastal vessels at berth (Table 3). The ranges for
EFs of $CO_2$, PM, TVOC, $SO_2$ in our study similar to those reported by Cooper et al. (2003), but our EFs of CO were much
higher and our EFs of $NO_x$ instead were much lower.
River vessels sail in inland rivers and many studies had investigated the emission from river vessels under crusing condition
(Fu et al., 2013; Peng et al., 2016; Zhang et al., 2016), but no studies are available about their emissions at berth. In this study,
river vessels used diesel as fuel, and they were not affected by the fuel switch policy. As showed in Table 3, for the tested river
vessels (ships H, I, J and K), the EFs of $CO_2$ (3014 ±99.0g kg$^{-1}$) and $NO_x$ (28.1 ±24.5 g kg$^{-1}$) were close to those for coastal
vessels; the EF of CO (77.9 ±62.5 g kg$^{-1}$), however, was nearly 4 times larger than that of coastal vessels after implementing
the fuel switch policy, and also larger than that reported for engineering vessel and research vessels under crusing condition
with the maximum of 30.2 g kg$^{-1}$ (Zhang et al., 2016); their EF of $SO_2$ was as low as 0.69 ±0.36 g kg$^{-1}$, while the EF of
NMHCs was as high as 3.36 ±2.77 g kg$^{-1}$, 85.6% larger than that for coastal vessels after implementing the policy, but fell in
the range for research vessels (1.24-4.18 g kg$^{-1}$) as reported by Zhang et al. (2016).
**3.2 EFs of grouped and individual NMHCs**
There are very sparse data about the EFs of grouped and individual NMHCs (Cooper et al., 1996; Murphy et al., 2010; Agrawal
et al., 2008; 2010), especially for ship emissions at berth. In this study, 68 species of NMHCs, including 29 alkanes, 21 alkenes,
1 alkyne and 17 aromatics, were determined. As showed in Fig. 3 and Table 4, for coastal vessels before implementing the
fuel switch policy, alkanes dominated the emissions among NMHCs with a share of 49.4 ±24.1% and an EF of 66.0 ±48.3
mg kg$^{-1}$, while aromatics and alkenes accounted for 27.9 ±12.3% and 21.9 ±11.9% of NMHCs with EFs of 29.2 ±8.6 mg kg$^{-1}$
and 21.9 ± 4.5 mg kg$^{-1}$, respectively. However, there were dramatic changes in the compositions of NMHCs after
implementing the fuel switch policy. Alkenes overtook alkanes to become the most abundant group with a share of 43.1% ±
12.8% and an EF of 924.6 ±1314.9 mg kg$^{-1}$, followed by alkanes (33.0 ±17.5%, 339.2 ±176.6 mg kg$^{-1}$) and aromatics (16.1
±4.1%, 247.3 ±236.4 mg kg$^{-1}$).
As for EFs of individual NMHCs, the top 25 species remain unchanged after the implementation of the fuel switch policy,
but their rankings have changed (Table S1). As showed in Fig. 4 and Table 4, n-undecane and n-dodecane were still among
the dominant species, although their percentages decreased a lot. Their EFs did not change as much, which were 22.5 ±18.2
mg kg$^{-1}$ and 21.5 ±17.1 mg kg$^{-1}$ before and 22.5 ±24.6 mg kg$^{-1}$ and 32.1 ±62.1 mg kg$^{-1}$, after the new policy, respectively.
Instead, EF of isobutane increased from 0.06 ±0.07 mg kg$^{-1}$ to 94.3 ±62.2 mg kg$^{-1}$. Striking increase in EFs was also observed
for alkenes. Ethylene overtook 1-hexene to become the most abundant alkene, with its EF increasing from 2.8 mg kg$^{-1}$ to 602
mg kg$^{-1}$ on average. Propene, with the EF of 5.5 ±1.5 mg kg$^{-1}$ before the fuel switch policy, had the second largest EF of 198
±260 mg kg$^{-1}$ after the fuel switch, an increase of over 30 times. 1-Hexene, which ranked the first among alkenes with the EF
of 5.9 ±3.8 mg kg$^{-1}$ before the fuel switch policy, also increased by 1.9 times to 17.3 ±19.4 mg kg$^{-1}$. The mass percentages of
acetylene, the only alkynes detected, increased from 0.9 ±0.6% to 7.5 ±7.6%, with its EF rose from 0.9 ±0.6 mg kg$^{-1}$ to 328.7
±605.4 mg kg$^{-1}$. Benzene and toluene were dominant aromatic species before and after the new policy. Their EFs increased





from 11.9 ±4.6 mg kg$^{-1}$ and 6.0 ±1.2 mg kg$^{-1}$ before to 116.5 ±200.8 mg kg$^{-1}$ and 33.3 ±42.5 mg kg$^{-1}$ after implementing the
policy, respectively.
The composition of NMHCs from river vessels were similar to that of coastal vessels after implementing the fuel switch
policy. As showed in Fig. 3 and Table S1, alkenes also dominated the emission of NMHCs with a share of 45.1 ±5.9%, while
aromatics and alkenes accounted for 33.7 ±4.8% and 14.3 ±4.1%, respectively. For individual NMHCs, the most abundant
species also were ethylene, isobutene, propene, acetylene, n-decane and benzene. However, the EFs of NMHCs for river
vessels were 1.9 times that of coastal vessels after implementing the fuel switch policy (Table 2), suggesting that VOCs
emissions from river vessels might played an important role as their emission are closer to populated areas and thus should be
regulated.
Very recently both Xiao et al. (2018) and Huang et al. (2018a) had carried out VOCs emission tests on ships at berth in
China's ECA. Xiao et al. (2018) reported that aromatics and alkanes dominated the VOCs emission from ships at berth.
Furthermore, the most abundant alkane species were n-heptane, methylcyclohexane, n-octane, n-nonane, n-decane and n-
undecane, and benzene and toluene contributed 9% of the VOCs emissions; Huang et al. (2018a) also investigated the VOCs
emission from ship at berth, but aromatics had a share up to 70.9%, while alkenes only accounted for 6.7%. The variety of the
ship fuels might be one of the key reasons for the big differences in compositions of VOC emissions among the available
studies. The new policy only restricted the FSC below 0.5%, so many types of fuels could be used in ships, as can be seen
from the four types of diesels by the tested ships (Fig. S1). Nonetheless, engine designs, performance and loads during sampling
might also lead to the differences (Cooper et al., 1996).
**3.3 Ozone and SOA formation potentials**
**3.3.1 OFPs of VOCs from ship exhausts**
Ozone Formation Potentials (OFPs) is the approach that uses maximum incremental reactivity (MIR) to represent the
maximum contribution of VOCs to near-surface ozone formation under optimal conditions (Carter, 2009). With ships emission
data in this study, the normalized ozone reactivity, ($R_{O3}$, g O$_3$ g$^{-1}$ VOCs) and OFPs (g O$_3$ kg$^{-1}$ Fuel) can be calculated as:
$R_{O3} = \sum_i w_i \times (\text{MIR})_i$ ,                                                                                                     (4)
$\text{OFPs} = \sum_i \text{EF}_i \times (\text{MIR})_i$ ,                                                                                             (5)
where $w_i$ is the mass percentage of total VOCs emissions for $i$ species.
As described in Fig. 5, the $R_{O3}$ of tested coastal vessels increased by nearly 70% from 3.19 ±0.82 g O$_3$ g$^{-1}$ VOCs to 5.41 ±
0.69 g O$_3$ g$^{-1}$ VOCs. The main reason for the rise of $R_{O3}$ is that shares of highly reactive alkenes (like ethylene and propene)
increased among the VOCs emitted, and the contribution percentages of alkenes to $R_{O3}$ increased from 56.4% ±13.3% to 75.7%
±13.3%. OFPs increased 28.7 times from 0.35 ±0.11 g O$_3$ kg$^{-1}$ Fuel to 10.37 ±13.55 g O$_3$ kg$^{-1}$ Fuel.
For river vessels, its average $R_{O3}$ was 5.55 g O$_3$ g$^{-1}$ VOCs, which was closed to that of coastal vessels after implementing
the fuel switch policy, but their average OFPs (22.98 ±16.59 g O$_3$ kg$^{-1}$ Fuel) was more than double that of coastal vessels. As





showed in Fig. S4, the $R_{O3}$ (4.22 g $O_3$ g$^{-1}$ VOCs) reported by Huang et al. (2018a) for ship emission after implementing the
fuel switch policy was about 20% lower than the $R_{O3}$ (5.41 g $O_3$ g$^{-1}$ VOCs) from this study, and the $R_{O3}$ of 2.63 $O_3$ g$^{-1}$ VOCs
reported by Xiao et al. (2018) is even lower than the $R_{O3}$ before implementing the policy in this study. These results also
suggests that there is great diversity in ship-emitted VOCs even in different regions of China.

### 3.3.2 SOAFPs of VOCs from ship exhausts

Similarly, normalized secondary organic aerosols (SOA) and SOA formation potentials (SOAFPs) can also be calculated as:
$R_{SOA} = \sum_i w_i \times Y_i$ ,                 (6)
$SOAFPs = \sum_i EF_i \times Y_i$ ,                 (7)
where $R_{SOA}$ is the normalized SOA reactivity (g SOA g$^{-1}$ VOCs); $Y_i$ is the SOA yield of VOC species $i$. Like Zhang et al.
(2018), we could calculate the SOAFPs under high-$NO_x$ and low-$NO_x$ conditions (Ng et al., 2007). However, we should be
cautious to interpret the results because intermediate volatile organic compounds (IVOCs) were not measured in this study,
and this may lead to underestimate SOA yields (Huang et al., 2018b; Lou et al., 2019).

254       As showed in Fig. 5, under high-$NO_x$ conditions, $R_{SOA}$ decreased by ~75% from 0.29 ±0.11 g SOA g$^{-1}$ VOCs to 0.07 ±0.08

g SOA g$^{-1}$ VOCs, while under low-$NO_x$ conditions $R_{SOA}$ also decreased by 66.5% from 0.31 ±0.09 g SOA g$^{-1}$ VOCs to 0.11
±0.09 g SOA g$^{-1}$ VOCs. This decline in $R_{SOA}$ was resulted from the decrease in mass percentages of aromatics and alkanes,
which have higher SOA yields than alkenes (Ng et al., 2007; Lim and Ziemann, 2009; Loza et al., 2014). However, with the
dramatically increased EFs of VOCs, under high-$NO_x$ conditions SOAFPs increased 1.6 times from 0.04 ±0.03 g SOA kg$^{-1}$
Fuel to 0.10 ±0.09 g SOA kg$^{-1}$ Fuel, and under low-$NO_x$ conditions SOAFPs increased 2.5 times from 0.04 ±0.03 g SOA kg$^{-1}$
Fuel to 0.14 ±0.11 g SOA kg$^{-1}$ Fuel.

261       In particular, the $R_{SOA}$ for ship F (Fig. S4) was significantly higher largely due to a higher fraction (11.5%) of n-dodecane,

which has the highest SOA yield among the NMHCs. For river vessels, the $R_{SOA}$ was the lowest in test ships, with the value
of 0.04 ±0.02 g SOA g$^{-1}$ VOCs under high-$NO_x$ conditions and 0.07 ±0.03 g SOA g$^{-1}$ VOCs under low-$NO_x$ conditions.
However, their SOAFPs was 0.17 ±0.13 g SOA kg$^{-1}$ Fuel under high-$NO_x$ conditions and 0.32 ±0.27 g SOA kg$^{-1}$ Fuel under
low-$NO_x$ conditions, which were instead the largest due to much higher EFs.

266       As showed in Fig. S4, Huang et al. (2018a) reported $R_{SOA}$ of 0.08 g SOA g$^{-1}$ VOCs under high-$NO_x$ conditions and 0.23 g

SOA g$^{-1}$ VOCs under low-$NO_x$ conditions. The higher $R_{SOA}$ are related to the higher fractions of aromatics in the VOC
emissions. Xiao et al. (2018) also reported an average $R_{SOA}$ of 0.02 g SOA g$^{-1}$ VOCs under high-$NO_x$ conditions, which was
even lower than the $R_{SOA}$ for river vessels in our study.

### 3.4 Conclusions

Ships emission control is primarily targeted on PM-related pollution and designating ECA with fuel switch policy is a widely
adopted approach to combat air pollution in harbor cities. In the present study, we measured emissions from coastal vessels at
berth in Guangzhou Port in the PRD region, one the three ECAs newly established since 2017, and preliminarily investigated





the changes in emissions caused by the fuel switch policy, and further compared the results with that measured for river vessels
unaffected by the new policy.
As reported by previous studies, our study also demonstrated that after implementing the fuel switch policy, the EFs of both
$SO_2$ and $PM_{2.5}$ for coastal vessels decreased, as evidenced by the fact that the EFs of $SO_2$ reduced by ~78.0% and the EFs of
$PM_{2.5}$ reduced by ~55.5% on average. However, the EF of VOCs increased about 14 times from 118 ±56.1 mg/kg to 1807 ±
1746 mg/kg. Moreover, the compositions of VOCs emitted from the coastal vessels also changed greatly. The mass percentages
of alkenes increased from 8.5%-27.3% to 44.4%-86.6%. The sharp increase of EFs, as well as elevated fractions of more
reactive species, resulted in much higher OFPs for VOCs emitted per kilogram fuel burned, which sharply increased about 29
times from 0.35 ±0.11 g $O_3$ $kg^{-1}$ Fuel to 10.37 ±13.55g $O_3$ $kg^{-1}$ Fuel. The SOAFPs also increased by over 50% although their
$R_{SOA}$ reduced by 66.5%-74.8%.
For river vessels unaffected by the fuel switch policy, the EFs of NMHCs were measured as high as 3358 ±2771 mg $kg^{-1}$,
nearly doubled those for coastal vessels after implementing the new policy, with OFPs and SOAFPs also about 2 times of their
counterparts for coastal vessels after implementing the policy.
In summary, our tests in the Guangzhou port demonstrated that for coastal vessels at berth, the fuel switch from high-sulfur
residual fuel oil to low-sulfur diesel or heavy oil did bring about largely decreased emissions of $SO_2$ and $PM_{2.5}$ and therefore
would benefit PM pollution control. However, the new policy raised another concern for the dramatic increase in emissions of
reactive VOCs from coastal vessels. This phenomenon is also reinforced by the fact that river vessels, which use diesel oils all
along and thus not affected by the fuel switch policy, also had much higher emissions of reactive VOCs. This larger emission
of reactive VOCs would probably worsen the ozone pollution and SOA formation in the harbor cities, how to further lower
the emission of reactive VOCs from ocean-going, coastal and river vessels is another regulatory and technological concern.
**Data availability**
The data used in this publication are available to the community and can be accessed by request to the corresponding author.
**Author contributions**
ZFW performed data analysis with contributions from YLZ and XMW. JJH, XLH, XY and WQY helped sampling. HZC and
YJW helped project coordinating and data interpretation. RQZ, MZ, HF and ZZ helped sample analysis.
**Competing interests**
The authors declare that they have no conflict of interest.



## Acknowledgements

This study was funded by Natural Science Foundation of China (41571130031/41530641), the National Key Research and Development Program (2016YFC0202204/2017YFC0212802), the Chinese Academy of Sciences (QYZDJ-SSW-DQC032/XDA23010303), Guangdong Science and Technology Department (2017BT01Z134/2016TQ03Z993), the Guangzhou Science Technology and Innovation Commission (201607020002), and Youth Innovation Promotion Association, CAS (2017406).

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



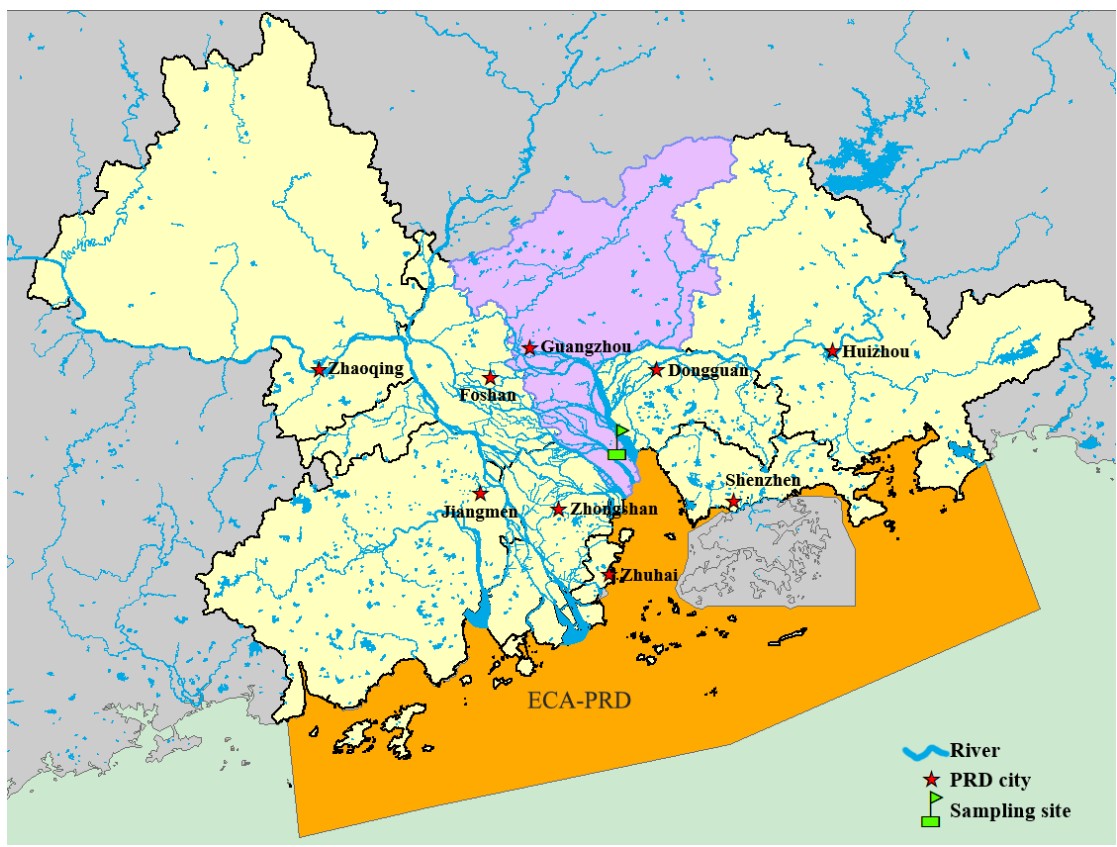


Figure 1. The realm of ECA-PRD and the sampling site.





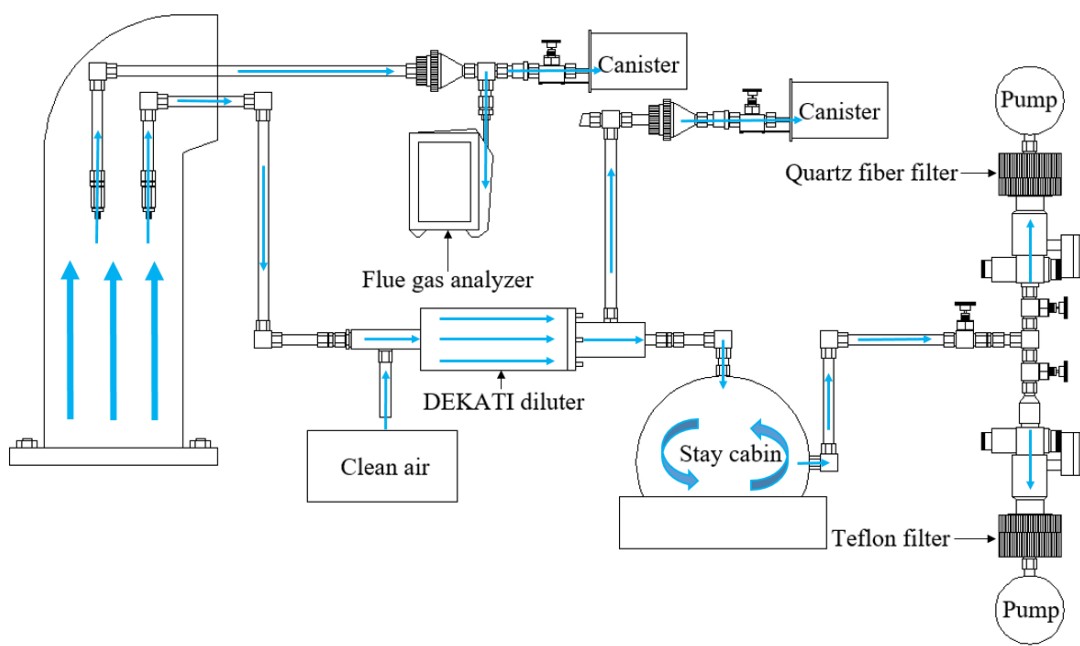


Figure 2. Schematic diagrams of sampling setup

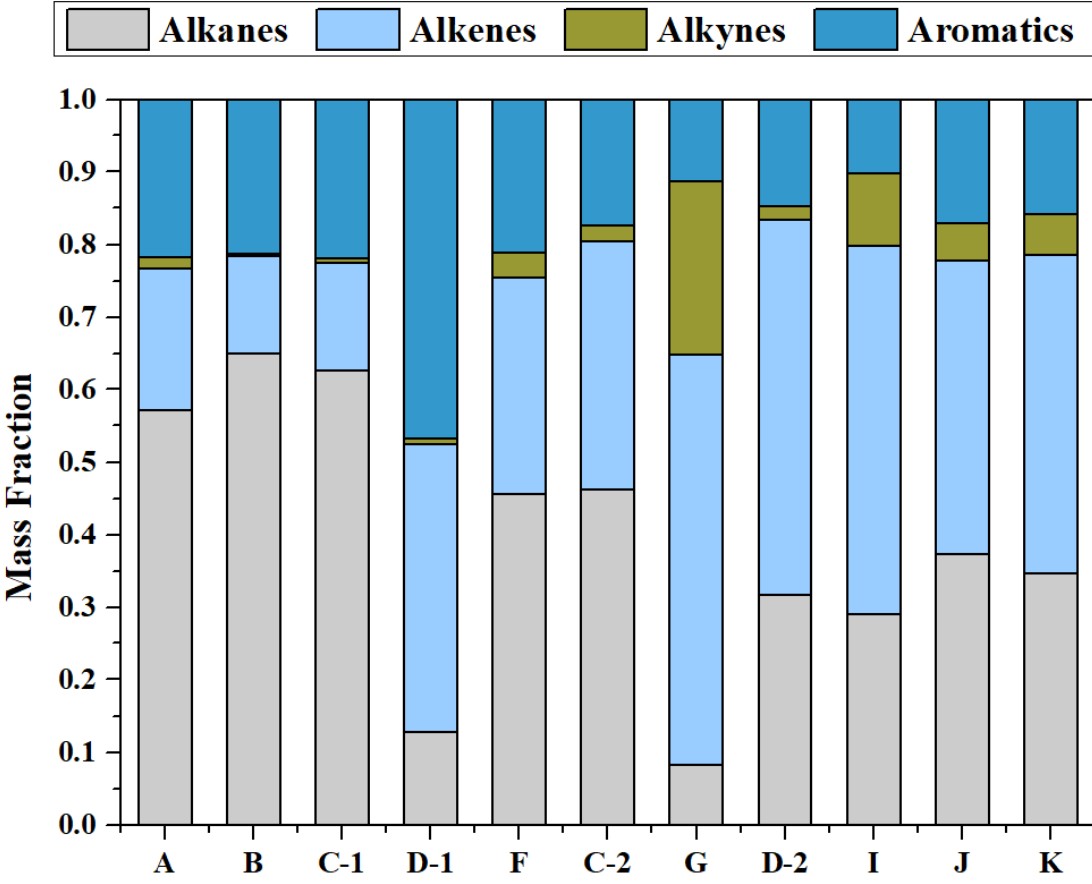

Figure 3. VOCs grouping according to their functional group. A, B, C-1 and D-1 are costal vessels tested before implementing the fuel switch policy, F, G, C-2 and D-2 are coastal vessels tested after implementing the fuel switch policy, and I, J and K are river vessels tested.





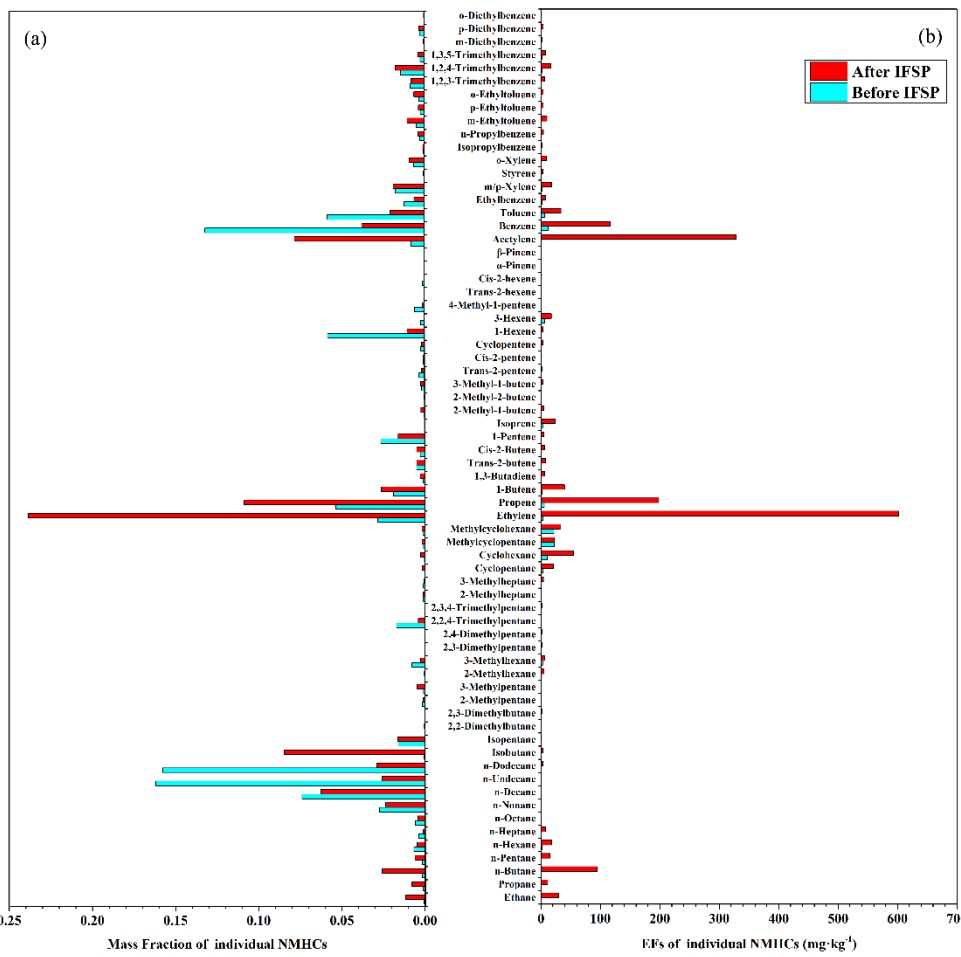


Figure 4. Comparison of VOCs emission factors before and after implementing the fuel switch policy (IFSP) for coastal vessels.





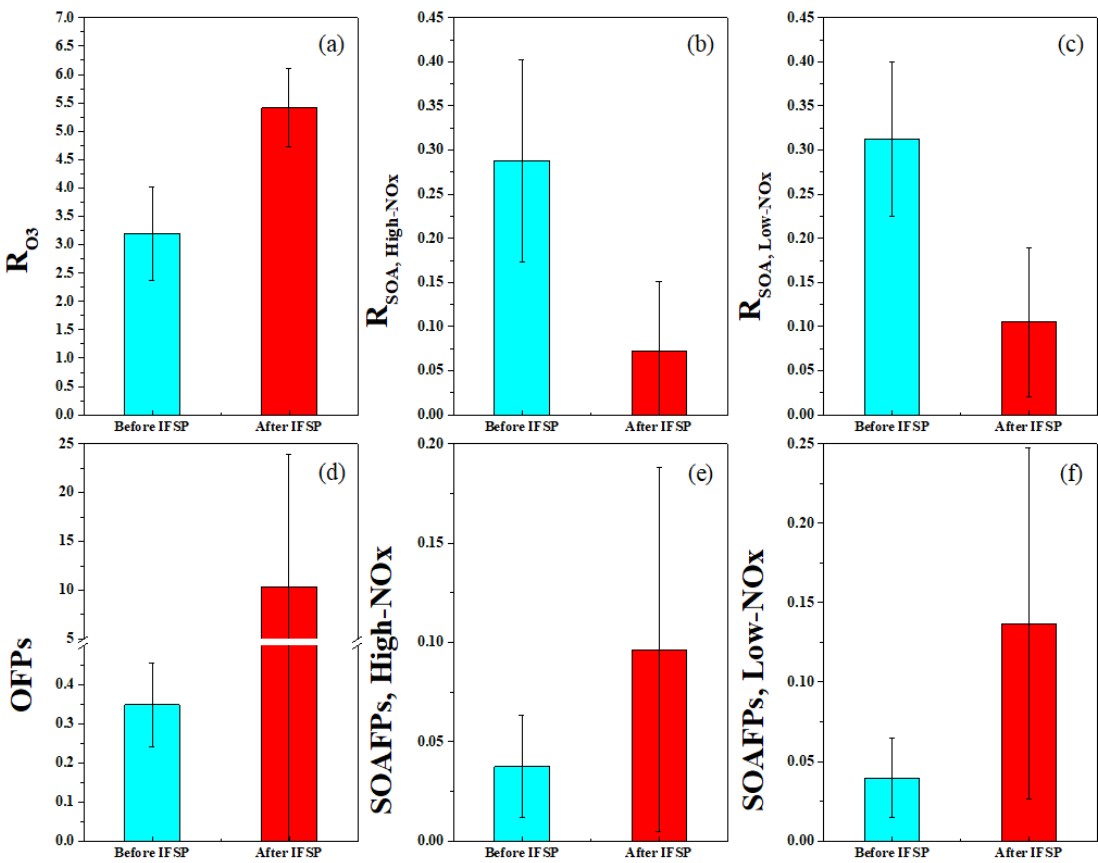


Figure 5. The changes in $R_{O3}$ (g $O_3$ g$^{-1}$ VOCs), $R_{SOA}$ (g SOA g$^{-1}$ VOCs), OFPs (g $O_3$ kg$^{-1}$ Fuel) and SOAFPs (g SOA kg$^{-1}$ Fuel)

for coastal vessels before and IFSP.





Table 1. The basic information of test vessels.

| NO | Test date | Ship types | Gross tonnage (t) | Vessel age (yr) | Auxiliary engine | | Fuel types | | |
|---|---|---|---|---|---|---|---|---|---|
| | | | | | Power (kW) | Amount | Types | C/% | S/% |
| Coastal vessels (before implementing the fuel switch policy) | | | | | | | | | |
| A | 2015.12.17 | container vessel | 47917 | 3 | 1760 / 1320 | 2 / 1 | residual oil | 84.9 | 1.60 |
| B | 2016.08.19 | container vessel | 41482 | 8 | 2045 | 3 | residual oil | 82.9 | 2.90 |
| C-1 | 2016.08.19 | container vessel | 49437 | 4 | 1760 / 1320 | 2 / 1 | residual oil | 82.7 | 2.10 |
| D-1 | 2016.11.15 | bulk carrier | 38384 | 3 | 660 | 3 | residual oil | 84.4 | 2.20 |
| Coastal vessels (after implementing the fuel switch policy) | | | | | | | | | |
| E | 2017.03.29 | bulk carrier | 8376 | 8 | 200 | 2 | diesel oil | 86.6 | 0.68 |
| F | 2017.12.22 | bulk carrier | 10716 | 10 | 200 | 3 | diesel oil | 86.6 | 0.13 |
| C-2 | 2018.04.21 | container vessel | 49437 | 6 | 1760 / 1320 | 2 / 1 | diesel oil | 85.8 | <0.01 |
| G | 2018.05.03 | container vessel | 25719 | 19 | 500 | 3 | heavy oil (low-sulfur) | 86.5 | 1.14 |
| D-2 | 2018.05.06 | bulk carrier | 38384 | 4 | 660 | 3 | heavy oil (low-sulfur) | 87.5 | 0.47 |
| River vessels | | | | | | | | | |
| H | 2017.03.29 | dry cargo carrier | 2445 | 9 | 144 / 76 | 2 / 1 | diesel oil | 86.0 | 0.06 |
| I | 2017.09.27 | container vessel | 1862 | 7 | 73.5 | 2 | diesel oil | 86.0 | 0.03 |
| J | 2017.09.27 | container vessel | 1357 | 15 | 58 | 2 | diesel oil | 86.1 | 0.03 |
| K | 2017.09.27 | container vessel | 1420 | 10 | 58.5 | 2 | diesel oil | 85.9 | 0.02 |






Table 2. The emission factors for test vessels (in unit of g kg$^{-1}$ fuel).

| Ships | $CO_2$ | CO | $SO_2$ | $NO_x$ | NMHCs | OC | EC | $PM_{2.5}$ |
|---|---|---|---|---|---|---|---|---|
| Coastal vessels (before implementing the fuel switch policy) | | | | | | | | |
| A | 3097 | 8.03 | 32.0 | 61.7 | 0.11 | 0.59 | 0.15 | 2.30 |
| B | 3029 | 5.33 | 58.0 | 29.1 | 0.20 | 0.29 | 0.05 | 1.46 |
| C-1 | 3025 | 3.80 | 42.0 | 19.9 | 0.11 | 0.22 | 0.07 | 1.02 |
| D-1 | 3069 | 14.6 | 44.0 | 51.5 | 0.06 | 0.16 | 0.61 | 2.44 |
| Coastal vessels (after implementing the fuel switch policy) | | | | | | | | |
| E | 3120 | 24.2 | 13.5 | 56.6 | 1.68 | 1.41 | 2.08 | 8.46 |
| F | 3156 | 5.50 | 2.52 | 13.0 | 1.11 | 0.55 | 1.41 | 2.17 |
| C-2 | 3130 | 6.16 | 0.06 | 19.0 | 0.71 | 0.16 | 0.29 | 0.56 |
| G | 3079 | 41.0 | 22.8 | 19.2 | 5.25 | 2.05 | 1.49 | 5.90 |
| D-2 | 3196 | 6.41 | 9.40 | 31.1 | 0.29 | 0.07 | 0.22 | 0.87 |
| River vessels | | | | | | | | |
| H | 3087 | 26.2 | 1.20 | 25.0 | 0.81 | 0.74 | 5.21 | 12.5 |
| I | 3120 | 24.2 | 13.5 | 56.6 | 1.68 | 1.41 | 2.08 | 8.46 |
| I | 3055 | 59.6 | 0.52 | 13.3 | 1.40 | - | - | - |
| J | 2865 | 171 | 0.68 | 9.77 | 6.93 | - | - | - |
| K | 3050 | 55.0 | 0.36 | 64.4 | 4.29 | - | - | - |




Table 3. Fuel-based average EFs (g kg⁻¹) from this study in comparison with those reported previously.

| Ships | FSC | Condition | $CO_2$ | CO | PM | TVOC | $SO_2$ | $NO_x$ |
|---|---|---|---|---|---|---|---|---|
| Coastal vessels or ocean-going vessels | | | | | | | | |
| Coastal vessels-Before IFSP[a,e] | >0.5% | At berth | 3055 | 7.93 | 1.81 | 0.12 | 44.0 | 40.6 |
| Coastal vessels-After IFSP[a,e] | <0.5% | At berth | 3136 | 16.7 | 3.59 | 1.81 | 9.66 | 27.8 |
| Passenger ferry-α[b] | 0.08% | At berth | 3080-3297 | 2.69-4.58 | 0.99-2.12 | 0.57-0.99 | 1.56-1.65 | 70.3-90.6 |
| Passenger ferry-β-1[b] | 0.53% | At berth | 3121-3284 | 4.34-6.99 | 1.96 | 0.29-0.57 | 10.2-11.0 | 54.4-71.6 |
| Passenger ferry-β-2[b] | 0.09% | At berth | 3200 | - | 1.29 | 1.71 | 1.67 | 84.2 |
| Passenger ferry-γ[b] | 1.20% | At berth | 3125-3226 | 1.50-2.60 | 1.37-2.00 | 0.87-1.14 | 23.7-24.1 | 64.7-84.7 |
| Car/truck carrier[b] | 0.23% | At berth | 3237-3251 | 4.31-4.59 | 0.80-0.89 | 0.89-1.08 | 4.68 | 45.0-46.4 |
| Container/ro-ro[b] | 2.20% | At berth | 3199-3212 | 3.55-4.17 | 2.49-3.10 | 0.79-0.88 | 44.0-44.2 | 59.4-70.4 |
| Chemical tanker[b] | 0.06% | At berth | 3159 | 3.22-3.41 | 0.65-0.75 | 1.36-1.40 | 1.21 | 81.8-83.6 |
| PanaMax Class Container[c] | 3.01% | Cruising | 2805 | 1.32 | 10.9 | - | 52.40 | 89.9 |
| River vessels | | | | | | | | |
| River vessels[a] | <0.5% | At berth | 3134 | 77.9 | 12.5 | 3.36 | 0.69 | 28.1 |
| Engineering vessel[d] | 0.08% | Cruising | 3071 | 30.2 | 9.40 | 23.7 | 1.60 | 115 |
| Research vessel-α[d] | 0.05% | Cruising | 3153 | 6.93 | 0.72 | 1.24 | 0.92 | 35.7 |
| Research vessel-β[d] | 0.13% | Cruising | 3151 | 9.20 | 0.16 | 4.18 | 2.60 | 31.6 |

[a]This study; [b]Cooper et al. (2003); [c]Agrawal et al. (2010); [d]Zhang et al. (2016); [e]implementing the fuel switch policy; [f]Zhang et al., (2018) with a coefficient
of 0.22 kg/kWh to convert g/kWh to g/kg.



Table 4. Emission factors (mg kg$^{-1}$) of NMHCs for test vessels.

| Species | Coastal vessels (before IFSP$^e$) | | | | Coastal vessels (after IFSP) | | | | River vessels | | |
|---|---|---|---|---|---|---|---|---|---|---|---|
| | A | B | C-1 | D-1 | F | C-2 | G | D-2 | I | J | K |
| Ethane | 0.1 | 0.1 | 0.1 | 0.1 | 8.8 | 5.6 | 99.0 | 3.4 | 17.4 | 59.4 | 31.6 |
| Propane | 0.1 | 0.1 | 0.1 | 0.1 | 14.6 | 3.6 | 24.5 | 2.7 | 2.4 | 9.0 | 7.5 |
| n-Butane | 0.3 | 0.1 | 0.4 | 0.0 | 5.6 | 20.7 | 15.4 | 19.3 | 0.6 | 2.1 | 149.3 |
| n-Hexane | 0.4 | 1.7 | 1.0 | 0.4 | 5.0 | 1.4 | 2.8 | 3.6 | 0.3 | 3.6 | 0.6 |
| n-Octane | 0.8 | 1.0 | 0.7 | 0.3 | 9.6 | 4.5 | 1.2 | 0.7 | 4.9 | 57.7 | 26.3 |
| n-Nonane | 4.6 | 4.5 | 4.1 | 0.3 | 43.0 | 37.3 | 1.4 | 0.9 | 20.5 | 199.6 | 144.5 |
| n-Decane | 2.4 | 23.2 | 15.2 | 0.8 | 117.3 | 97.9 | 2.2 | 1.7 | 32.8 | 300.5 | 247.5 |
| n-Undecane | 21.0 | 45.7 | 22.9 | 0.3 | 45.6 | 42.8 | 0.7 | 0.7 | 24.7 | 195.9 | 179.9 |
| n-Dodecane | 26.8 | 42.5 | 15.5 | 1.3 | 127.2 | 1.0 | 0.2 | 0.1 | 0.7 | 6.8 | 57.6 |
| Isobutane | 0.2 | 0.04 | 0.04 | ND | 88.5 | 73.3 | 180.0 | 35.2 | 252.1 | 1336.5 | 459.1 |
| Isopentane | 2.2 | 1.1 | 2.0 | 1.2 | 14.5 | 14.1 | 35.6 | 7.6 | 23.6 | 171.3 | 73.4 |
| 3-Methylhexane | 0.8 | 1.0 | 1.5 | 0.3 | 3.1 | 1.4 | 15.6 | 1.0 | 7.0 | 36.8 | 35.0 |
| TM224PE$^a$ | ND | 4.1 | 1.3 | 2.2 | 2.8 | 4.0 | 18.0 | 1.4 | 9.0 | 73.5 | 32.8 |
| Other alkanes | 1.8 | 3.0 | 1.8 | 0.6 | 21.2 | 18.4 | 34.6 | 14.4 | 11.1 | 129.0 | 43.2 |
| Sum of alkanes | 61.5 | 128.3 | 66.5 | 7.8 | 506.8 | 326.2 | 431.1 | 92.7 | 407.1 | 2581.9 | 1488.4 |
| Ethylene | 2.9 | 3.2 | 2.2 | 3.1 | 170.5 | 96.7 | 2062.7 | 79.3 | 401.8 | 1155.1 | 1125.2 |
| Propene | 7.1 | 6.3 | 3.7 | 4.9 | 82.8 | 71.1 | 595.2 | 42.8 | 201.1 | 969.5 | 378.3 |
| 1-Butene | 2.1 | 0.6 | 2.6 | 1.7 | 23.9 | 21.1 | 102.7 | 10.1 | 32.0 | 149.0 | 105.6 |
| Trans-2-butene | 0.6 | 0.4 | 0.5 | 0.5 | 3.9 | 5.5 | 17.6 | 1.7 | 5.7 | 34.0 | 21.0 |
| 1-Pentene | 4.1 | 2.0 | 1.2 | 2.9 | 17.3 | 14.7 | 57.9 | 5.2 | 24.7 | 143.1 | 80.4 |
| 1-Hexene | 2.5 | 10.3 | 2.8 | 8.1 | 7.9 | 11.1 | 46.6 | 3.5 | 18.0 | 127.1 | 68.9 |
| M4PE1ENE$^b$ | 0.7 | 1.1 | 0.3 | 0.7 | 1.4 | 1.5 | 10.4 | 0.6 | 3.0 | 26.4 | 12.6 |
| Other alkenes | 1.1 | 2.7 | 2.4 | 2.2 | 23.1 | 19.9 | 82.5 | 7.2 | 26.1 | 206.0 | 96.8 |
| Sum of alkenes | 21.1 | 26.5 | 15.8 | 24.0 | 330.8 | 241.6 | 2975.6 | 150.4 | 712.5 | 2810.3 | 1888.8 |
| Acetylene | 1.8 | 0.7 | 0.6 | 0.5 | 38.5 | 15.4 | 1255.1 | 5.6 | 139.1 | 355.5 | 241.8 |
| Benzene | 9.6 | 11.6 | 7.9 | 18.6 | 18.3 | 13.0 | 423.7 | 10.9 | 46.6 | 191.7 | 129.5 |
| Toluene | 5.4 | 7.6 | 4.8 | 6.3 | 15.7 | 7.8 | 98.2 | 11.7 | 22.1 | 131.3 | 75.5 |
| Ethylbenzene | 1.1 | 2.5 | 1.8 | 0.7 | 7.4 | 5.3 | 13.1 | 3.0 | 6.3 | 61.5 | 28.2 |
| m/p-Xylene | 1.8 | 3.5 | 1.7 | 1.3 | 24.1 | 19.4 | 20.4 | 7.0 | 11.5 | 129.1 | 57.4 |
| o-Xylene | 0.6 | 1.5 | 0.7 | 0.5 | 14.1 | 10.1 | 9.3 | 2.9 | 6.3 | 69.1 | 31.6 |
| m-Ethyltoluene | 0.7 | 1.5 | 0.5 | 0.2 | 24.8 | 11.4 | 2.0 | 1.4 | 8.4 | 100.0 | 75.9 |
| o-Ethyltoluene | 0.3 | 1.2 | 0.6 | 0.1 | 16.8 | 6.1 | 1.7 | 0.9 | 5.0 | 54.2 | 28.9 |
| TM123B$^c$ | 1.1 | 2.4 | 1.1 | 0.2 | 19.7 | 9.5 | 2.2 | 0.8 | 5.5 | 71.1 | 43.9 |
| TM124B$^d$ | 1.0 | 5.3 | 2.1 | 0.2 | 44.1 | 18.3 | 3.3 | 1.6 | 15.2 | 167.8 | 99.7 |
| Other aromatics | 1.7 | 4.6 | 2.2 | 0.3 | 49.1 | 21.6 | 15.5 | 2.8 | 15.5 | 206.2 | 105.2 |
| Sum of aromatics | 23.3 | 41.8 | 23.2 | 28.4 | 234.0 | 122.6 | 589.5 | 43.0 | 142.5 | 1182.0 | 675.7 |

$^a$2,2,4-Trimethylpentane; $^b$4-Methyl-1-pentene; $^c$1,2,3-Trimethylbenzene; $^d$1,2,4-Trimethylbenzene.