# Peer review of "Dramatic increase in reactive VOC emissions from ships at berth after implementing the fuel switch policy in the Pearl River Delta Emissions Control Area"

_Atmospheric Chemistry and Physics, 2019_

## Referee Comment (RC1) · Anonymous Referee #1 · 25 Nov 2019

Ship emissions as important sources of air pollution at the coastal cities have raised widespread attention and their emission characteristics have been consistently studied by many researchers. Wu et al. presents the changes of VOC emissions from ships at berth after implementing the fuel switch policy at the ECA. They find that the apparent increase of reactive species in the VOC emissions due to the strategy and their second formation potentials including O3 and SOA are also estimated. This study is well motivated for the effect evaluation of emission control strategies. However, despite the potential meaning of results from this study, the presentation of this study needs be

improved to a large extent, especially for the writing. Major comments: 1. Description of VOCs. The author measured 68 VOC species used by GC-MS/FID, but the author is very chaotic for the description of VOC species in this manuscript, using the term NMHCs or VOCs in different sentences. Which one is the accurate expression? Generally, NMHC concentrations are determined by subtracting the amount of $CH_4$ constituents from the THC measured by FID. The samples collected in canisters and analyzed by a preconcentrator coupled to GC-MSD/FID are speciated VOCs. Could the PEMS system measure THC and $CH_4$ concentrations? 2. More information about sampling and analysis is needed, such as sampling flow, sampling time, sampling temperature, the auxiliary load, the devices used for conventional pollutants, the and the standard gas for VOC measurement. 3. As mentioned by the author, the fuel composition is a very important factor for VOC profiles, which is a possible reason for the different VOC compositions of the tested ships with the previous results. Then, is there apparent difference of VOC compositions for tested ships using four different fuels? What is the trendy of the VOC emissions when correlating the diesel composition? 4. More concise. Academic writing is a big question for this manuscript. There are many simple mistakes appeared in substantial sentences, which are mostly summarized in minor comments. Polishing the language is strongly suggested. 5. The unified expression. The author wrote several types of phases to express the implementation of the fuel switch policy, such as after implementing the fuel switch policy, after the new policy, after implementing the policy, after the fuel switch, and after the implementation of the fuel switch policy. Choose a suitable phrase for this expression.

Minor comments: 1. Line 24 The unit of EF is not unitized, mg/kg and mg kg-1. 2. Line 26 more rich. . . is it not richer? 3. Line 34 The number of $PM_{2.5}$ should be subscripted. 4. Line 34 "may threatens"? It doesn't need the plural form for the term "threaten". 5. Line 46 ECAs? 6. Line 54 ship emissions? 7. Line 57 Is it suitable using the word "combat"? 8. Line 78 reveals? 9. Line 84 emissions from ships? 10. Line 99 Are all the pollutant emissions accounted for 40%? 11. Line 113 What is a $PM_{2.5}$ cutting head? Please give an accurate description. 12. Line 120 Is the mass selective detector MSD?

What about the mass spectrometer detector? 13. Line 105 have already used? 14. Line 133 The EF of CO2 is calculated not determined. As follows not as following. 15. Line 136 Is the unit of Cf (gÂůkg-1)ïij§ 16. Line 140 concentration? 17. Line 151-153 Why is the explanation of VOC composition change placed in this section? 18. Line 161 The number of C6 should be subscripted. 19. Line 156 What is NMHCs? Is the measured VOC species? 20. Line 158 limited... 21. Line 160 The EFs of CO2... 22. Line 160 Is it right "before to"? 23. Line 159-163 It should give a summary rather than displaying the tested results of every ship. 24. Line 166 The term "that" should be "those". 25. Line 168 What is the carbonaceous aerosol? Does that mean OC and EC? 26. Line 169 "As shown" is the correct form, please revise all of the forms in this manuscript. 27. Line 172 the EFs of? 28. Line 176 by marine gasoline? 29. Line 178 TVOCs? Does TVOCs denote the measured VOC species? 30. Line 180 the emissions? 31. Line 191 NMHCs? 32. Line 199 and 214 individual NMHCs? 33. Line 206 the fuel switch? 34. Line 208 the only alkynes? 35. Line 212 "were" should be revised to "was". 36. Line 217 might played? 37. Line 217 their emission are? 38. Line 223 emission from shipïij§ 39. Line 230 Ozone Formation Potentials (OFPs) is? 40. Line 245 ship-emitted VOCs at berth... 41. Line 247 Please give the literature for the calculated method of SOAFPs. 42. Line 247 normalized secondary organic aerosol reactivity? 43. Line 250 Like Zhang et.al reported? 44. Line 258-260 What is the problem told by the SOAFP difference under the high NOx and low NOx conditions? 45. Line 256 This decline of RSOA? 50. Line 262 What is the NMHCs? 51. Line 266-267 How about the comparison of Huang et al. results and this study results? 52. Line 268 What is the reason for the lower RSOA of Xiao's results? 53. Line 271 Ships? 54. Line 273 one the three? 55. Line 278 the EF of VOCïij§Line 281 the EF of NMHCs? Which one is right? 56. Line 281 Why explained the unit of fuel-based EF here? 57. Line 287 are not affected ? 58. Table 2 g kg-1 fuel? 59. Figure 5 before and IFSP? 60. Figure 4 The figure needs add the standard error bar. IFSP is the first appearance. Spell out all acronyms on first use in the abstract and in the body of the article. 61. Figure 2 diagrams?

---

## Referee Comment (RC2) · Anonymous Referee #2 · 26 Nov 2019

The manuscript of Wu et al. with the title of "Dramatic increase of reactive VOC emission from ships at berth after implementing the fuel switch policy in the Pearl River Delta Emission Control Area" characterized the emissions of PM2.5, SO2 and VOCs from ships at berth before and after implementing the fuel switch policy with a FSC limit of 0.5% in the Pearl River Delta ECA in south China. After implementing the fuel switch policy, the EFs of SO2 and PM2.5 for coastal vessels dropped 23 by 78% and 56%, however, the EFs of NMHCs increased by a factor of 15 times before implementing the new policy. The reactive alkenes overtook alkanes to become the dominant group,

which led to the sharp increase of ozone formation potential. The results showed that this change may threatens ozone pollution control in the harbor cities. This is a well-written manuscript. The results could help to improve our understanding of another side of fuel switch policy and raised the concerns for reactive VOCs emissions from ships, and found river vessels might had even larger emissions of VOCs and NOx than coastal vessels. I highly recommend this manuscript to be accepted by the journal.

Minor revisions,

line 18, change "to reduce" to for reducing";

line 19, change "matters" to "matter";

line 19, change "emission" to "emissions";

line 20, change "south" to "southern";

line 21, change "a" to "an";

line 21, delete "in south China";

line 22, change "emission" to "emissions";

line 22, change "coastal vessels" to "the coastal vessels";

line 22, change "dropped" to "decreased";

line 23, change "the EFs of non-methane hydrocarbons (NMHCs), however, reached" to "however, the EFs of the nonmethane hydrocarbons (NMHCs), were";

line 24, change "about" to "approximately";

line 24, change "emission" to "emissions";

line 25, change "NMHCs" to "the NMHCs";

line 26, change "more rich" to "richer";

line 28, change "the new policy" to "the new policy was implemented";

line 28, change "reactive alkenes" to "the reactive alkenes";

line 29, change "for per kilogram of fuel burned, emitted NMHCs" to "the emitted NMHCs per kg of fuel burned";

line 29, change "about" to "approximately";

line 30, change "coastal vessels" to "the coastal vessels";

line 30, change "river vessels" to "the river vessels";

line 31, change "their EFs of NMHCs" to "the EFs of their NMHCs";

line 35, change "coastal or ocean-going vessels" to "the coastal or ocean-going vessels";

line 35, change "along with" to "in addition to";

line 35, change "river vessels" to "the river vessels";

line 38, change "total world merchandise trade" to "the total global merchandise trade";

line 39, change "it is no surprise that" to "unsurprisingly";

line 42, change "would give rise to" to "can cause";

line 42, change "areas," to "areas";

line 42, change "environmental burden" to "the environmental burden";

line 46, change "more" to "a more";

line 47, change "brought about" to "resulted in";

line 49, change "resulted" to "has resulted";

line 50, change "aerosols" to "aerosols, respectively,"

line 51, change "revealed" to "has revealed";

line 55, change "reduce" to "be reduced";

line 56, change "emission sources" to "emissions sources";

line 67, change "VOCs" to "the VOCs";

line 68, change "north" to "northern";

line 69, change "As a matter of fact, previous" to "Previous";

line 70, change "Meanwhile" to "In addition";

line 75, change "January 1, 2017 to December 31, 2019" to "1 January, 2017, to 31 December, 2019";

line 76, change "estimiated" to "estimated";

line 76, change "atmosheric" to "atmospheric";

line 77, change "south" to "southern";

line 81, change "bring about" to "result in";

line 82, change "shut down" to "shut down,";

line 82, change "emission" to "emissions";

line 88, change "like" to "such as";

line 92, change "in comparison" to "for a comparison";

line 97, change "the fifth" to "fifth";

line 98, change "the seventh" to "seventh";

line 98, change "contribute near 40%" to "account for nearly 40% of";

line 99, change "nine" to "the nine";

line 101, change "FSC" to "the FSC";

line 101, change "be lower" to "have been less";

line 102, change "that" to "in that";

line 147, change "on average decreased from 2.2 $\pm$ 0.5%" to "decreased from 2.2 $\pm$ 0.5% on average";

line 148, change "though" to "although";

line 150, change "hydrocarbons" to "hydrocarbons,";

line 151, change "chromatograms" to "chromatograms, than those of coastal vessels before the policy";

line 157, change "performance of combustion system" to "the performance of the combustion system";

line 185, change ", and also larger" to "and higher";

line 185, change "engineering vessel" to "engineering vessels";

line 185, change "crusing condition" to "cruising conditions";

line 186, change "the maximum" to "a maximum";

line 199, change "individual" to "the individual";

line 199, change "remain" to "remained";

line 210, change "before" to "before implementing the new policy";

line 210, change "after implementing the policy, respectively" to ", respectively, after implementing the policy";

line 212, change "NMHCs from river vessels were" to "the NMHCs from the river vessels was";
line 212, change "coastal vessels" to "the coastal vessels";

line 213, change "showed" to "shown";

line 213, change "dominated the emission of NMHCs with a share of" to "were dominant in the emissions of the NMHCs at";

line 214, change "individual" to "the individual";

line 223, change "had a share up to 70.9%" to "accounted for up to 70.9% of those emissions";

line 223, change "variety of the" to "variation in";

line 224, change "big" to "large";

line 224, change "compositions of VOC emissions" to "the compositions the VOC emissions";

line 225, change "only restricted" to "restricted only";

line 225, delete "can be";

line 226, change "diesels" to "diesel fuels used";

line 226, change "sampling might also lead" to "the sampling might have also led";

line 228, change "potentials" to "potential";

line 229, change "OFPs of VOCs from ship exhausts" to "OFP of the VOCs from ship exhaust";

line 230, change "Formation Potentials (OFPs)" to "formation potentials (OFP)";

line 231, change "ships emission" to "ships emissions";

line 232, change "reactivity," to "reactivity";

line 237, change "rise of" to "increase in";

[Figure]

line 237, change "like" to "such as";

line 240, change "coastal vessels" to "the coastal vessels";

line 241, change "OFPs" to "OFP";

line 241, change "coastal vessels" to "the coastal vessels";

line 245, change "suggests" to "suggest";

line 246, change "SOAFPs of VOCs from ship exhausts" to "SOAFP of the VOCs from ship exhaust";

line 250, change "Like" to "Similar to";

line 251, change "SOAFPs" to "SOAFP";

line 252, change "to interpret" to "in interpreting";

line 252, delete "(IVOCs)";

line 261, change "higher" to "higher than that of the other ships,";

line 262, change "has: to "had";

line 262, change "river vessels" to "the river vessels";

line 262, change "test ships" to "the tested ships";

line 262, change the value" to "a value";

line 273, change "one the three ECAs newly established" to "one of the three newly established ECAs";

line 280, change "of EFs" to "in the EFs";

line 280, change "more" to "the more";

line 288, change "did bring about largely" to "resulted in substantially";

line 289, change "for" to "due to";

line 290, change "coastal vessels" to "the coastal vessels";

line 290, change "is" to "was";

line 290, change "use" to "had used";

line 290, change "all along and thus" to "the entire time and thus were";

line 291, change "much higher" to "high";

line 291, change "larger emission" to "high level of emissions";

line 292, delete "would";

line 292, change "how to further lower the emission" to "and further lowering the emissions";

---

## Author Comment (AC1) · 5 Dec 2019

**Response to the reviewer's comments**

**"Dramatic increase in reactive VOC emissions from ships at berth after**

**implementing the fuel switch policy in the Pearl River Delta Emissions Control**

**Area"** *by* **Zhen-Feng Wu et al.**

**Reviewer #1**

Ship emissions as important sources of air pollution at the coastal cities have raise widespread attention and their emission characteristics have been consistently studied by many researchers. Wu et al.

presents the changes of VOC emissions from ships at berth after implementing the fuel switch policy at the ECA. They find that the apparent increase of reactive species in the VOC emissions due to the strategy and their second formation potentials including O3 and SOA are also estimated. This study is well motivated for the effect evaluation of emission control strategies. However, despite the potential meaning of results from this study, the presentation of this study needs be improved to a large extent, especially for the writing.

**Major Comments**

1. Description of VOCs. The author measured 68 VOC species used by GC-MS/FID, but the author is very chaotic for the description of VOC species in this manuscript, using the term NMHCs or VOCs in different sentences. Which one is the accurate expression? Generally, NMHC concentrations are determined by subtracting the amount of CH4 constituents from the THC measured by FID. The samples collected in canisters and analyzed by a preconcentrator coupled to GC-MSD/FID are speciated VOCs. Could the PEMS system measure THC and CH4 concentrations?

Reply: Thanks. Yes, generally NMHC concentrations are determined by subtracting the amount of $CH_4$

constituents from the THC measured by FID. In this study, we collected samples in canisters and measured 68 VOC species by a preconcentrator coupled to the GC-MSD/FID. We further measured

$CH_4$ in the canister samples by a gas chromatograph (Agilent 6980GC, USA) with a flame ionization detector and a packed column (5A molecular sieve 60/80 mesh, 3m $\times$ 1/8 in.). We did not report $CH_4$ in the manuscript since our concern is focused on photochemically reactive species. As the 68 VOC

species we determined are C2-C12 hydrocarbons, sometimes we just used the term "NMHC" when referring to the 68 VOCs in our manuscript. To avoid confusion, in the revised manuscript we have replaced all "NMHCs" with "VOCs".

2. More information about sampling and analysis is needed, such as sampling flow, sampling time, sampling temperature, the auxiliary load, the devices used for conventional pollutants, and the standard gas for VOC measurement.

Reply: Thanks. As suggested, in the revised manuscript we have added more information about sampling flow, sampling time, devices used for conventional pollutants and standard gas for VOCs measurement as below:

"The ship exhaust first entered a Dekati® ejector dilutor (DI-1000, Dekati Ltd., Finland) from the sampling nozzle and then was spilt into four parts after being diluted with clean air: one part was for air sampling with 2 L canisters and 4 L Teflon bags for 3-5 min after passing through a filter; two other parts were for collecting $PM_{2.5}$ samples with 47 mm Teflon filters (Whateman, Mainstone, UK) and 47 mm quartz fiber filters (Whateman, Mainstone, UK), respectively, at a flow of 16.7 L min$^{-1}$ for 20-30 min, after the diluted exhaust was mixed well in a stay cabin, and then passing through a $PM_{2.5}$ separator; and the last part was the vent. Before dilution, the concentrations of $CO_2$, CO, $SO_2$ and $NO_x$ in the ship exhaust were directly measured by a flue gas analyzer (F-550, WOHLER, Germany) while air samples were also collected simultaneously by a 2L canisters and a 4L Teflon bags." (lines 120-127 in the revised manuscript).

We have added more information about the VOCs standards. "The calibration standards were prepared by dynamically diluting the 100 ppbv Photochemical Assessment Monitoring Stations (PAMS) standard mixture (57 NMHCs including 15 AHs) and TO-14 standard mixture (39 compounds) from Spectra Gases Inc., NJ, USA to 0.5, 1, 5, 15 and 30 ppbv. More details about the analysis are described elsewhere ( Zhang et al., 2013; 2015; Yang et al., 2018)." Line 131-138.

Besides, we have also added sampling temperature and the auxiliary load in Table S1 in the supporting information as showed below:

Table S1. More information during sampling.

| NO | Sampling | Auxiliary engine |
| --- | --- | --- |

|  | temperature (℃) | Power(kW) | Amount | Condition | Engine loads (%) | Fuel consumption rate(t*d⁻¹) |
|---|---|---|---|---|---|---|
| | | Coastal vessels (before IFSP) | | | | |
| A | 17 | 1760 | 2 | Off | - | - |
| | | 1320 | 1 | On | 53 | 3.0 |
| B | 32 | 2045 | 2 | Off | - | - |
| | | 2045 | 1 | On | 40 | 4.1 |
| C-1 | 34 | 1760 | 2 | Off | - | - |
| | | 1320 | 1 | On | 55 | 4.0 |
| D-1 | 29 | 660 | 1 | Off | - | - |
| | | 660 | 2 | On | 34 | 2.2 |
| | | Coastal vessels (after IFSP) | | | | |
| E | 25 | 200 | 1 | Off | - | - |
| | | 200 | 1 | On | 39 | 0.4 |
| F | 21 | 200 | 2 | Off | - | - |
| | | 200 | 1 | On | 50 | 0.5 |
| C-2 | 29 | 1760 | 2 | Off | - | - |
| | | 1320 | 1 | On | 52 | 3.5 |
| G | 31 | 500 | 2 | Off | - | - |
| | | 500 | 1 | On | 65 | 1.8 |
| D-2 | 31 | 660 | 1 | Off | - | - |
| | | 660 | 2 | On | 37 | 2.4 |
| | | River vessels | | | | |
| H | 25 | 76 | 1 | Off | - | - |
| | | 144 | 1 | Off | - | - |
| | | 144 | 1 | On | 40 | 0.3 |
| I | 32 | 73.5 | 2 | On | 40 | 0.3 |
| J | 38 | 58 | 1 | Off | - | - |
| | | 58 | 1 | On | 32 | 0.1 |
| K | 35 | 58.8 | 1 | Off | - | - |
| | | 58.8 | 1 | On | 35 | 0.1 |

Reference

Yang, W. Q., Zhang, Y. L., Wang, X. M., Li, S., Zhu, M., Yu, Q. Q., Li, G. H., Huang, Z. H., Zhang, H. N., Wu, Z. F., Song, W., Tan, J. H., and Shao, M.: Volatile organic compounds at a rural site in Beijing: influence of temporary emission control and wintertime heating, Atmos. Chem. Phys., 18, 12663-12682, https://doi.org/10.5194/acp-18-12663-2018, 2018.

Zhang, Y. L., Wang, X. M., Barletta, B., Simpson, I. J., Blake, D. R., Fu, X. X., Zhang, Z., He, Q. F., Liu, T. Y., Zhao, X. Y., and Ding, X.: Source attributions of hazardous aromatic hydrocarbons in urban, suburban and rural areas in the Pearl River Delta (PRD) region, J. Hazard. Mater., 250, 403-411, https://doi.org/10.1016/j.jhazmat.2013.02.023, 2013.

Zhang, Y. L., Wang, X. M., Zhang, Z., Lv, S. J., Huang, Z. H., and Li, L. F.: Sources of $C_2$-$C_4$ alkenes, the most important ozone nonmethane hydrocarbon precursors in the Pearl River Delta region, Sci.

Total Environ., 502, 236-245, https://doi.org/10.1016/j.scitotenv.2014.09.024, 2015.

3. As mentioned by the author, the fuel composition is a very important factor for VOC profiles, which is a possible reason for the different VOC compositions of the tested ships with the previous results.

Then, is there apparent difference of VOC compositions for tested ships using four different fuels?

What is the trendy of the VOC emissions when correlating the diesel composition?

Reply: We simply measured solvent-extractable fraction of the oils by GC-MSD as some fuels are very sticky residue oils before the fuel switch policy. Nonetheless, as showed in Figure S1, we could see that after the implementing the fuel switch policy, there is a tendency to have more factions of low molecular weight hydrocarbons (or hydrocarbons having lower carbon numbers). As for ship C, the residual oil used before the fuel switch policy was mainly composed of saturated $C_{11}$-$C_{36}$ alkanes; after implementing the new policy, however, the residue oil used by ship C was replaced with diesel oil with no peaks after heptacosane (C27) in its total ion chromatographs. For ship D, before implementing the fuel switch policy it used residual oil slightly different from that used by ships A, B and C in its compositions, particularly in relative high fractions of naphthalene and methylnaphthalenes apart from saturated alkanes. After implementing the fuel switch policy, ship D instead used low-sulfur heavy oil.

Although the responses of the most hydrocarbons did not change very much, the responses of low carbon number species, including naphthalene, tridecanes and methylnaphthalenes, became relatively higher, and lower carbon number species such as indene (C8) were also detected. As a result, we found the mass percentages of $< C_6$ VOCs (VOCs with carbon numbers below 6) in the total VOCs in ship exhaust increased from 8.5%-27.3% to 44.4%-86.6% after implementing the fuel switch policy. As described in the manuscript, we noticed that the fuel used by the ships became more abundant in low molecular weight fractions, but we did not conduct a comprehensive analysis of the fuel compositions and we do not know if the fuels we samples are representative enough, so we feel it would be inappropriate to go further saying more in this aspect. As a matter of fact, after we report our results to local administrations, they determined to start a full-scale survey about fuels used by ships.

[Figure]

          Figure S1. Typical total ion chromatographs of VOC species in fuel oils.

4. More concise. Academic writing is a big question for this manuscript. There are many simple mistakes appeared in substantial sentences, which are mostly summarized in minor comments. Polishing the language is strongly suggested.

Reply: Really sorry for making so many simple mistakes. Thanks a lot for your hard work in carefully checking the manuscript. We have also requested an academic editing service "SPRINGER NATURE Author Services (SNAS)" to improve the English language, grammar, punctuation, spelling, and overall style by one or more of the highly qualified native English speaking editors at SNAS. The verification code is 04C9-9B0B-7E9B-561C-1839.

5. The unified expression. The author wrote several types of phases to express the implementation of the fuel switch policy, such as after implementing the fuel switch policy, after the new policy, after implementing the policy, after the fuel switch, and after the implementation of the fuel switch policy. Choose a suitable phrase for this expression.

Reply: Thanks for the suggestion. In the revised manuscript, we use "the implementation of the fuel switch policy" to unify the expression.

**Minor Comments**

1. Line 24 The unit of EF is not unitized, mg/kg and mg kg-1.

Reply: As suggested, we have unitized the unit of EF in mg kg$^{-1}$.

2. Line 26 more rich… is it not richer?

Reply: As suggested, we have replaced "more rich" with "richer".

3. Line 34 The number of PM2.5 should be subscripted.

Reply: As suggested, we have replaced "PM2.5" with "PM$_{2.5}$".

4. Line 34 "may threatens"? It doesn't need the plural form for the term "threaten".

Reply: As suggested, we have replaced "threatens" with "threaten".

5. Line 46 ECAs?

Reply: As suggested, we have replaced "ECA" with "ECAs".

6. Line 54 ship emissions?

Reply: As suggested, we have replaced "ship emission" with "ship emissions".

7. Line 57 Is it suitable using the word "combat"?

Reply: As suggested, we have replaced "combat" with "control".

8. Line 78 reveals?

Reply: As suggested, we have replaced "reveal" with "reveals".

9. Line 84 emissions from ships?

Reply: As suggested, we have replaced "emissions from ship" with "emissions from ships".

10. Line 99 Are all the pollutant emissions accounted for 40%?

Reply: Yes, all the pollutant emissions, including $SO_2$, $NO_x$, CO, $PM_{10}$, $PM_{2.5}$ and VOCs accounted for nearly 40%, as shown below (Li et al., 2016a):

[Figure]

a) Ports

Reference

Li, C., Yuan, Z. B., Ou, J. M., Fan, X. L., Ye, S. Q., Xiao, T., Shi, Y. Q., Huang, Z. J., Ng, S. K. W.,

Zhong, Z. M., and Zheng, J. Y.: An AIS-based high-resolution ship emission inventory and its uncertainty in Pearl River Delta region, China, Sci. Total Environ., 573, 1-10, https://doi.org/10.1016/j.scitotenv.2016.07.219, 2016a.

11. Line 113 What is a PM2.5 cutting head? Please give an accurate description.

Reply: As suggested, we have replaced "$PM_{2.5}$ cutting head" with "$PM_{2.5}$ separator".

12. Line 120 Is the mass selective detector MSD? What about the mass spectrometer detector?

Reply: Here MSD represents mass selective detector. Both mass selective detector and mass spectrometer detector are often abbreviated as MSD.

13. Line 105 have already used?

Reply: As suggested, we have replaced "already used" with "have already used".

14. Line 133 The EF of CO2 is calculated not determined. As follows not as following.

Reply: As suggested, we have replaced "determined" with "calculated".

15. Line 136 Is the unit of Cf (g kg-1)

Reply: It is $C_F$ instead of Cf. In the revised manuscript, we have changed the expression in line 154 as

"$C_F$ is the carbon content per kg of fuel (g kg$^{-1}$);"

16. Line 140 concentration?

Reply: As suggested, we have replaced "concentrations" with "concentration".

17. Line 151-153 Why is the explanation of VOC composition change placed in this section?

Reply: As suggested, we have moved this part to line 200-202.

18. Line 161 The number of C6 should be subscripted.

Reply: As suggested, we have replaced "C6" with "$C_6$".

19. Line 156 What is NMHCs? Is the measured VOC species?

Reply: Yes, it refers to the measured VOC species. We have replaced "NMHCs" with "VOCs".

20. Line 158 limited…

Reply: As suggested, we have replaced "limit" with "limited".

21. Line 160 The EFs of CO2

Reply: As suggested, we have replaced "The EF of $CO_2$" with "The EFs of $CO_2$".

22. Line 160 Is it right "before to"?

Reply: We have deleted "before".

23. Line 159-163 It should give a summary rather than displaying the tested results of every ship.

Reply: Because ships C and D were tested both before and after the implementation of the fuel switch policy, the changes in emissions for the two ships would be more convincing in reflecting the influence of the fuel switch policy. This is why we particularly display the tested results of ships C and D.

24. Line 166 The term "that" should be "those".

Reply: As suggested, we have replaced "that" with "those".

25. Line 168 What is the carbonaceous aerosol? Does that mean OC and EC?

Reply: Yes, carbonaceous aerosol included OC and EC.

26. Line 169 "As shown" is the correct form, please revise all of the forms in this manuscript.

Reply: As suggested, we have replaced "As showed" with "As shown" in the whole manuscript.

27. Line 172 the EFs of?

Reply: As suggested, we have replaced "the EF of" with "the EFs of".

28. Line 176 by marine gasoline?

Reply: The "marine gasoil" was mentioned in Copper et al. (2003) and it referred to a kind of diesel.

Reference

Cooper, D. A.: Exhaust emissions from ships at berth, Atmos. Environ., 37, 3817-3830, https://doi.org/10.1016/S1352-2310(03)00446-1, 2003.

29. Line 178 TVOCs? Does TVOCs denote the measured VOC species?

Reply: Yes, TVOCs denoted the total measured VOC species.

30. Line 180 the emissions?

Reply: As suggested, we have replaced "emission" with "emissions".

31. Line 191 NMHCs?

Reply: As suggested, we have replaced "NMHCs" with "VOCs".

32. Line 199 and 214 individual NMHCs?

Reply: As suggested, we have replaced "NMHCs" with "VOCs".

33. Line 206 the fuel switch?

Reply: We have replaced "after the fuel switch" with "after the implementation of the fuel switch policy".

34. Line 208 the only alkynes?

Reply: Yes, we only measured acetylene in this study.

35. Line 212 "were" should be revised to "was".

Reply: As suggested, we have replaced "were" with "was".

36. Line 217 might played?

Reply: We have replaced "might played" with "might play".

37. Line 217 their emission are?

Reply: We have replaced "their emission are" with "their emissions are".

38. Line 223 emission from ship

Reply: We have replaced "emission from ship" with "emissions from ships".

39. Line 230 Ozone Formation Potentials (OFPs) is?

Reply: We have replaced "Ozone Formation Potentials (OFPs)" with "Ozone formation potential
(OFP)".

40. Line 245 ship-emitted VOCs at berth…

Reply: As suggested, we have replaced "ship-emitted VOCs" with "ship-emitted VOCs at berth".

41. Line 247 Please give the literature for the calculated method of SOAFPs.

Reply: As suggested, we added "(Zhang et al., 2018a)" in line 283.

Reference

Zhang, Y. L., Yang, W. Q., Simpson, I., Huang, X. Y., Yu, J. Z., Huang, Z. H., Wang, Z. Y., Zhang, Z.,
Liu, D., Huang, Z. Z., Wang, Y. J., Pei, C. L., Shao, M., Blake, D. R., Zheng, J. Y., Huang, Z. J., and
Wang, X. M.: Decadal changes in emissions of volatile organic compounds (VOCs) from on-road
vehicles with intensified automobile pollution control: Case study in a busy urban tunnel in south
China, Environ. Pollut., 233, 806-819, https://doi.org/10.1016/j.envpol.2017.10.133, 2018a.

42. Line 247 normalized secondary organic aerosol reactivity?

Reply: As suggested, we have replaced "normalized secondary organic aerosols (SOA)" with
"normalized secondary organic aerosol reactivity ($R_{SOA}$, g SOA g$^{-1}$ VOCs)".

43. Line 250 Like Zhang et.al reported?

Reply: We deleted "Like Zhang et.al reported".

44. Line 258-260 What is the problem told by the SOAFP difference under the high NOx and low NOx conditions?

Reply: In this method, $Y_i$ is the SOA yield of VOC species i, as determined by chamber studies (Ng et al., 2007; Lim and Ziemann, 2009; Loza et al., 2014). SOA yields of VOCs depend on nitrogen oxide ($NO_x$) (Ng et al., 2007). Thus, we calculated the SOAFPs under high-NOx and low-NOx conditions, respectively.

Reference

Lim, Y. B., and Ziemann, P. J.: Effects of molecular structure on aerosol yields from OH

radical-initiated reactions of linear, branched, and cyclic alkanes in the presence of NOx, Environ. Sci.

Technol., 43, 2328-2334, https://doi.org/10.1021/es803389s, 2009.

Loza, C. L., Craven, J. S., Yee, L. D., Coggon, M. M., Schwantes, R. H., Shiraiwa, M., Zhang, X.,

Schilling, K. A., Ng, N. L., Canagaratna, M. R., Ziemann, P. J., Flagan, R. C., and Seinfeld, J. H.:

Secondary organic aerosol yields of 12-carbon alkanes, Atmos. Chem. Phys., 14, 1423-1439, https://doi.org/10.5194/acp-14-1423-2014, 2014.

Ng, N. L., Kroll, J. H., Chan, A. W. H., Chhabra, P. S., Flagan, R. C., and Seinfeld, J. H.: Secondary organic aerosol formation from m-xylene, toluene, and benzene, Atmos. Chem. Phys., 7, 3909-3922, https://doi.org/10.5194/acp-7-3909-2007, 2007.

45. Line 256 This decline of RSOA?

Reply: As suggested, we have replaced "This decline in RSOA" with "This decline of $R_{SOA}$".

50. Line 262 What is the NMHCs?

Reply: We have replaced "NMHCs" with "VOCs".

51. Line 266-267 How about the comparison of Huang et al. results and this study results?

Reply: Huang et al. (2018a) also measured the emissions of VOCs from ship at berth using low-sulfur fuels, so we could directly compare with the coastal vessels after the implementation of the fuel switch policy. We have changed the expression in line 304-307 as below:

"As shown in Fig. S4, based on the VOCs emissions from ship at berth reported in Huang et al.

(2018a), we calculated a RSOA of 0.080 g SOA g-1 VOCs under high-NOx conditions and 0.228 g

SOA g-1 VOCs under low-NOx conditions for a coastal vessel also using low-sulfur fuels. This relatively higher RSOA under low-NOx conditions was related to the higher fractions of aromatics in the VOC emissions."

52. Line 268 What is the reason for the lower RSOA of Xiao's results?

Reply: The reason for lower $R_{SOA}$ of Xiao's results was that they adopted another method in Gentner et al. (2012) using another set of SOA yield for hydrocarbons as shown in Table R1-1. In the revised manuscript we added explanations for this in line 307-309 "Using another method in Gentner et al.

(2012), Xiao et al. (2018) reported an average $R_{SOA}$ of 0.017 g SOA $g^{-1}$ VOCs under high-$NO_x$

conditions, which was close to a $R_{SOA}$ of 0.015 g SOA $g^{-1}$ VOCs calculated by the same method for the coastal vessels after IFSP.

Table R1-1. Average high-NOx SOA yields in Gentner et al. (2012)

| Carbon number | Straight-chain alkanes | Branched alkanes | Cycloalkanes (single straight alkyl chain) | Cycloalkanes (branched or multiple alkyl chain (s)) | Bicycloalkanes | Tricycloalkanes | Aromatics | Polycyclic aromatics compounds |
|---|---|---|---|---|---|---|---|---|
| 1 | - | - | - | - | - | - | - | - |
| 2 | - | - | - | - | - | - | - | - |
| 3 | - | - | - | - | - | - | - | - |
| 4 | - | - | - | - | - | - | - | - |
| 5 | - | - | - | - | - | - | - | - |

| | | | | | | | | |
|---|---|---|---|---|---|---|---|---|
| 6 | - | - | 0.0004 | - | - | - | 0.14 | - |
| 7 | - | - | 0.0007 | 0.0001 | - | - | 0.083 | - |
| 8 | 0.0006 | 0.0001 | 0.0015 | 0.0002 | - | - | 0.048 | - |
| 9 | 0.0012 | 0.0002 | 0.0031 | 0.0005 | 0.0005 | - | 0.077 | - |
| 10 | 0.0026 | 0.0004 | 0.0059 | 0.001 | 0.001 | - | 0.12 | 0.17 |
| 11 | 0.0053 | 0.0008 | 0.01 | 0.0018 | 0.0018 | - | 0.15 | 0.23 |
| 12 | 0.01 | 0.0017 | 0.016 | 0.0034 | 0.0031 | 0.0032 | 0.19 | 0.28 |
| 13 | 0.019 | 0.0035 | 0.026 | 0.0062 | 0.0056 | 0.0057 | 0.26 | 0.4 |
| 14 | 0.033 | 0.007 | 0.041 | 0.011 | 0.0097 | 0.0098 | 0.33 | 0.49 |
| 15 | 0.055 | 0.013 | 0.064 | 0.019 | 0.016 | 0.017 | 0.39 | 0.62 |
| 16 | 0.089 | 0.024 | 0.099 | 0.031 | 0.026 | 0.027 | 0.43 | 0.7 |
| 17 | 0.14 | 0.042 | 0.16 | 0.053 | 0.044 | 0.045 | 0.46 | 0.75 |
| 18 | 0.23 | 0.073 | 0.24 | 0.088 | 0.072 | 0.073 | 0.51 | 0.79 |
| 19 | 0.37 | 0.12 | 0.36 | 0.14 | 0.12 | 0.12 | 0.56 | 0.82 |
| 20 | 0.56 | 0.2 | 0.5 | 0.22 | 0.19 | 0.19 | 0.61 | 0.82 |
| 21 | 0.77 | 0.32 | 0.66 | 0.33 | 0.29 | 0.3 | 0.65 | 0.82 |
| 22 | 0.96 | 0.47 | 0.82 | 0.45 | 0.43 | 0.43 | 0.67 | 0.82 |
| 23 | 1.08 | 0.61 | 0.94 | 0.57 | 0.56 | 0.57 | 0.68 | 0.82 |
| 24 | 1.14 | 0.7 | 1.03 | 0.67 | 0.66 | 0.67 | 0.68 | 0.82 |
| 25 | 1.16 | 0.75 | 1.09 | 0.73 | 0.74 | 0.74 | 0.68 | 0.82 |

Reference

Gentner, D. R., Isaacman, G., Worton, D. R., Chan, A. W. H., Dallmann, T. R., Davis, L., Liu, S., Day,

D. A., Russell, L. M., Wilson, K. R., Weber, R., Guha, A., Harley, R. A., and Goldstein, A. H.:

Elucidating secondary organic aerosol from diesel and gasoline vehicles through detailed characterization of organic carbon emissions, Proc. Natl. Acad. Sci. U. S. A., 109, 18318-18323, https://doi.org/10.1073/pnas.1212272109, 2012.

53. Line 271 Ships?

Reply: We have replaced "Ships" with "Ship".

54. Line 273 one the three?

Reply: We have replaced "one the three" with "one of the three".

55. Line 278 the EF of VOCs and Line 281 the EF of NMHCs? Which one is right?

Reply: We have replaced "NMHCs" with "VOCs" in the whole manuscript.

56. Line 281 Why explained the unit of fuel-based EF here?

Reply: We have deleted "for VOCs emitted per kilogram fuel burned".

57. Line 287 are not affected?

Reply: We have changed "For river vessels unaffected" to "For the river vessels were not affected".

58. Table 2 g kg-1 fuel?

Reply: This shows the unit of the emission factor in the table.

59. Figure 5 before and IFSP?

Reply: We have changed "before and IFSP" as "before and after the implementation of the fuel switch policy".

60. Figure 4 The figure needs add the standard error bar. IFSP is the first appearance. Spell out all acronyms on first use in the abstract and in the body of the article.

Reply: Thanks for the suggestion. We have added the standard error bar in Figure 4 and spell out the

 acronyms of IFSP on its first use in the abstract and in the body of the article in line 21 and line 86.

[Figure]

Figure 4. Comparison of VOCs emission factors before and after IFSP for coastal vessels.

61. Figure 2 diagrams?

Reply: We have replaced "diagrams" with "diagram".

---

## Author Comment (AC2) · 5 Dec 2019

**Response to the reviewer's comments**

**"Dramatic increase in reactive VOC emissions from ships at berth after implementing the fuel switch policy in the Pearl River Delta Emissions Control Area"** *by* **Zhen-Feng Wu et al.**

**Reviewer #2**

The manuscript of Wu et al. with the title of "Dramatic increase of reactive VOC emission from ships at berth after implementing the fuel switch policy in the Pearl River Delta Emission Control Area" characterized the emissions of PM2.5, SO2 and VOCs from ships at berth before and after implementing the fuel switch policy with a FSC limit of 0.5% in the Pearl River Delta ECA in south China. After implementing the fuel switch policy, the EFs of SO2 and PM2.5 for coastal vessels dropped 23 by 78% and 56%, however, the EFs of NMHCs increased by a factor of 15 times before implementing the new policy. The reactive alkenes overtook alkanes to become the dominant group, which led to the sharp increase of ozone formation potential. The results showed that this change may threatens ozone pollution control in the harbor cities. This is a well-written manuscript. The results could help to improve our understanding of another side of fuel switch policy and raised the concerns for reactive VOCs emissions from ships, and found river vessels might had even larger emissions of VOCs and NOx than coastal vessels. I highly recommend this manuscript to be accepted by the journal.

**Minor revisions**

line 18, change "to reduce" to for reducing";

Reply: Revised as suggested, Line 18, replaced "to reduce" to "for reducing".

line 19, change "matters" to "matter";

Reply: Revised as suggested, Line 19, replaced "matters" to "matter".

line 19, change "emission" to "emissions";

Reply: Revised as suggested, Line 19, replaced "emission" to "emissions".

line 20, change "south" to "southern";

Reply: Revised as suggested, Line 21, replaced "south" to "southern".

line 21, change "a" to "an";

Reply: Revised as suggested, Line 21, replaced "a" to "an".

line 21, delete "in south China";

Reply: Revised as suggested, Line 22, deleted "in south China".

line 22, change "emission" to "emissions";

Reply: Revised as suggested, Line 23, replaced "emission" to "emissions".

line 22, change "coastal vessels" to "the coastal vessels";

Reply: Revised as suggested, Line 23, replaced "coastal vessels" to "the coastal vessels".

line 22, change "dropped" to "decreased";

Reply: Revised as suggested, Line 23, replaced "dropped" to "decreased".

line 23, change "the EFs of non-methane hydrocarbons (NMHCs), however, reached" to "however, the
EFs of the nonmethane hydrocarbons (NMHCs), were";

Reply: Revised as suggested, Line 24, replaced "the EFs of non-methane hydrocarbons (NMHCs),
however, reached to" to "however, the EFs of the VOCs were".

line 24, change "about" to "approximately";

Reply: Revised as suggested, Line 25, replaced "about" to "approximately".

line 24, change "emission" to "emissions";

Reply: Revised as suggested, Line 26, replaced "emission" to "emissions".

line 25, change "NMHCs" to "the NMHCs";

Reply: Revised as suggested, Line 26, replaced "NMHCs" to "the VOCs".

line 26, change "more rich" to "richer";

Reply: Revised as suggested, Line 27, replaced "more rich" to "richer".

line 28, change "the new policy" to "the new policy was implemented";

Reply: Revised as suggested, Line 29, replaced "the new policy" to "IFSP".

line 28, change "reactive alkenes" to "the reactive alkenes";

Reply: Revised as suggested, Line 30, replaced "reactive alkenes" to "the reactive alkenes".

line 29, change "for per kilogram of fuel burned, emitted NMHCs" to "the emitted NMHCs per kg of fuel burned";

Reply: Revised as suggested, Line 30, replaced "for per kilogram of fuel burned, emitted NMHCs" to

"the emitted VOCs per kg of fuel burned".

line 29, change "about" to "approximately";

Reply: Revised as suggested, Line 31, replaced "about" to "approximately".

line 30, change "coastal vessels" to "the coastal vessels";

Reply: Revised as suggested, Line 32, replaced "coastal vessels" to "the coastal vessels".

line 30, change "river vessels" to "the river vessels";

Reply: Revised as suggested, Line 32, replaced "river vessels" to "the river vessels".

line 31, change "their EFs of NMHCs" to "the EFs of their NMHCs";

Reply: Revised as suggested, Line 34, replaced "their EFs of NMHCs" to "the EFs of their VOCs".

line 35, change "coastal or ocean-going vessels" to "the coastal or ocean-going vessels";

Reply: Revised as suggested, Line 38, replaced "coastal or ocean-going vessels" to "the coastal or ocean-going vessels".

line 35, change "along with" to "in addition to";

Reply: Revised as suggested, Line 38, replaced "along with" to "in addition to".

line 35, change "river vessels" to "the river vessels";

Reply: Revised as suggested, Line 39, replaced "river vessels" to "the river vessels".

line 38, change "total world merchandise trade" to "the total global merchandise trade";

Reply: Revised as suggested, Line 42, replaced "total world merchandise" to "the total global merchandise trade".

line 39, change "it is no surprise that" to "unsurprisingly";

Reply: Revised as suggested, Line 44, replaced "it is no surprise that" to "unsurprisingly".

line 42, change "would give rise to" to "can cause";

Reply: Revised as suggested, Line 46, replaced "would give rise to" to "can cause".

line 42, change "areas," to "areas";

Reply: Revised as suggested, Line 46, replaced "areas," to "areas".

line 42, change "environmental burden" to "the environmental burden";

Reply: Revised as suggested, Line 47, replaced "environmental burden" to "the environmental burden".

line 46, change "more" to "a more";

Reply: Revised as suggested, Line 51, replaced "more" to "a more".

line 47, change "brought about" to "resulted in";

Reply: Revised as suggested, Line 52, replaced "brought about" to "resulted in".

line 49, change "resulted" to "has resulted";

Reply: Revised as suggested, Line 54, replaced "resulted" to "has resulted".

line 50, change "aerosols" to "aerosols, respectively,"

Reply: Revised as suggested, Line 55, replaced "aerosols" to "aerosols, respectively".

line 51, change "revealed" to "has revealed";

Reply: Revised as suggested, Line 56, replaced "revealed" to "has revealed".

line 55, change "reduce" to "be reduced";

Reply: Revised as suggested, Line 60, replaced "reduce" to "be reduced".

line 56, change "emission sources" to "emissions sources";

Reply: Revised as suggested, Line 62, replaced "emission sources" to "emissions sources".

line 67, change "VOCs" to "the VOCs";

Reply: Revised as suggested, Line 73, replaced "VOCs" to "the VOCs".

line 68, change "north" to "northern";

Reply: Revised as suggested, Line 74, replaced "north" to "northern".

line 69, change "As a matter of fact, previous" to "Previous";

Reply: Revised as suggested, Line 75, replaced "As a matter of fact, previous" to "Previous".

line 70, change "Meanwhile" to "In addition";

Reply: Revised as suggested, Line 77, replaced "Meanwhile" to "In addition".

line 75, change "January 1, 2017 to December 31, 2019" to "1 January, 2017, to 31 December, 2019";

Reply: Revised as suggested, Line 83, replaced "January 1, 2017 to December 31, 2019" to "1 January,

2017, to 31 December, 2019".

line 76, change "estimiated" to "estimated";

Reply: Revised as suggested, Line 83, replaced "estimiated" to "estimated".

line 76, change "atmosheric" to "atmospheric";

Reply: Revised as suggested, Line 84, replaced "atmosheric" to "atmospheric".

line 77, change "south" to "southern";

Reply: Revised as suggested, Line 84, replaced "south" to "southern".

line 81, change "bring about" to "result in";

Reply: Revised as suggested, Line 89, replaced "bring about" to "result in".

line 82, change "shut down" to "shut down,";

Reply: Revised as suggested, Line 90, replaced "shut down" to "shut down,".

line 82, change "emission" to "emissions";

Reply: Revised as suggested, Line 90, replaced "emission" to "emissions".

line 88, change "like" to "such as";

Reply: Revised as suggested, Line 96, replaced "like" to "such as".

line 92, change "in comparison" to "for a comparison";

Reply: Revised as suggested, Line 100, replaced "in comparison" to "for a comparison".

line 97, change "the fifth" to "fifth";

Reply: Revised as suggested, Line 106, replaced "the fifth" to "fifth".

line 98, change "the seventh" to "seventh";

Reply: Revised as suggested, Line 107, replaced "the seventh" to "seventh".

line 98, change "contribute near 40%" to "account for nearly 40% of";

Reply: Revised as suggested, Line 107, replaced "contribute near 40%" to "account for nearly 40% of".

line 99, change "nine" to "the nine";

Reply: Revised as suggested, Line 108, replaced "nine" to "the nine".

line 101, change "FSC" to "the FSC";

Reply: Revised as suggested, Line 111, replaced "FSC" to "the FSC".

line 101, change "be lower" to "have been less";

Reply: Revised as suggested, Line 111, replaced "be lower" to "have been less".

line 102, change "that" to "in that";

Reply: Revised as suggested, Line 112, replaced "that" to "in which".

line 147, change "on average decreased from 2.2 ± 0.5%" to "decreased from 2.2 ± 0.5% on average";

Reply: Revised as suggested, Line 167, replaced "on average decrease from 2.2 ± 0.5%" to "decreased
from 2.2 ± 0.5% on average".

line 148, change "though" to "although";

Reply: Revised as suggested, Line 168, replaced "though" to "although".

line 150, change "hydrocarbons" to "hydrocarbons,";

Reply: Revised as suggested, Line 171, replaced "hydrocarbons" to "hydrocarbons,".

line 151, change "chromatograms" to "chromatograms, than those of coastal vessels before the policy";

Reply: Revised as suggested, Line 171, replaced "chromatograms" to "chromatograms, than those of coastal vessels before IFSP".

line 157, change "performance of combustion system" to "the performance of the combustion system";

Reply: Revised as suggested, Line 178, replaced "performance of combustion system" to "the performance of the combustion system".

line 185, change ", and also larger" to "and higher";

Reply: Revised as suggested, Line 211, replaced "and also larger" to "and larger".

line 185, change "engineering vessel" to "engineering vessels";

Reply: Revised as suggested, Line 211, replaced "engineering vessel" to "engineering vessels".

line 185, change "crusing condition" to "cruising conditions";

Reply: Revised as suggested, Line 212, replaced "crusing condition" to "cruising conditions".

line 186, change "the maximum" to "a maximum";

Reply: Revised as suggested, Line 212, replaced "the maximum" to "a maximum".

line 199, change "individual" to "the individual";

Reply: Revised as suggested, Line 229, replaced "individual" to "the individual".

line 199, change "remain" to "remained";

Reply: Revised as suggested, Line 229, replaced "remain" to "remained".

line 210, change "before" to "before implementing the new policy";

Reply: Revised as suggested, Line 232, replaced "before" to "before IFSP".

line 242, change "after implementing the policy, respectively" to ", respectively, after implementing the policy";

Reply: Revised as suggested, Line 242, replaced "after implementing the policy, respectively" to

"respective, after IFSP".

line 212, change "NMHCs from river vessels were" to "the NMHCs from the river vessels was";

Reply: Revised as suggested, Line 243, replaced "NMHCs from river vessels were" to "the VOCs from the river vessels was".

line 212, change "coastal vessels" to "the coastal vessels";

Reply: Revised as suggested, Line 243, replaced "coastal vessels" to "the coastal vessels".

line 213, change "showed" to "shown";

Reply: Revised as suggested, Line 244, replaced "showed" to "shown".

line 213, change "dominated the emission of NMHCs with a share of" to "were dominant in the emissions of the NMHCs at";

Reply: Revised as suggested, Line 244, replaced "dominated the emission of NMHCs with a share of"

to "were dominant in the emissions of the NMHCs at".

line 214, change "individual" to "the individual";

Reply: Revised as suggested, Line 246, replaced "individual" to "the individual".

line 223, change "had a share up to 70.9%" to "accounted for up to 70.9% of those emissions";

Reply: Revised as suggested, Line 254, replaced "had a share up to 70.9%" to "accounted for up to

70.9% of those emissions".

line 223, change "variety of the" to "variation in";

Reply: Revised as suggested, Line 255, replaced "variety of the" to "variation in".

187 line 224, change "big" to "large";

188 Reply: Revised as suggested, Line 256, replaced "big" to "large".

189 line 224, change "compositions of VOC emissions" to "the compositions the VOC emissions";

190 Reply: Revised as suggested, Line 256, replaced "compositions of VOC emissions" to "the

191 compositions of the VOC emissions".

192 line 225, change "only restricted" to "restricted only";

193 Reply: Revised as suggested, Line 256, replaced "only restricted" to "restricted only".

194 line 225, delete "can be";

195 Reply: Revised as suggested, Line 257, deleted "can be".

196 line 226, change "diesels" to "diesel fuels used";

197 Reply: Revised as suggested, Line 258, replaced "diesels" to "diesel fuels used".

198 line 226, change "sampling might also lead" to "the sampling might have also led";

199 Reply: Revised as suggested, Line 258, replaced "sampling might also lead" to "the sampling might

200 have also led".

201 line 228, change "potentials" to "potential";

202 Reply: Revised as suggested, Line 260, replaced "potentials" to "potential".

203 line 229, change "OFPs of VOCs from ship exhausts" to "OFP of the VOCs from ship exhaust";

204 Reply: Revised as suggested, Line 261, replaced "OFPs of VOCs from ship exhausts" to "OFP of the

205 VOCs from ship exhaust".

206 line 230, change "Formation Potentials (OFPs)" to "formation potentials (OFP)";

207 Reply: Revised as suggested, Line 262, replaced "Formation Potentials (OFPs)" to "formation potential (OFP)".

line 231, change "ships emission" to "ships emissions";

Reply: Revised as suggested, Line 263, replaced "ships emission" to "ships emissions".

line 232, change "reactivity," to "reactivity";

Reply: Revised as suggested, Line 264, replaced "reactivity," to "reactivity".

line 237, change "rise of" to "increase in";

Reply: Revised as suggested, Line 270, replaced "rise of" to "increase in".

line 237, change "like" to "such as";

Reply: Revised as suggested, Line 271, replaced "like" to "such as".

line 240, change "coastal vessels" to "the coastal vessels";

Reply: Revised as suggested, Line 274, replaced "coastal vessels" to "the coastal vessels".

line 241, change "OFPs" to "OFP";

Reply: Revised as suggested, Line 275, replaced "OFPs" to "OFP".

line 241, change "coastal vessels" to "the coastal vessels";

Reply: Revised as suggested, Line 276, replaced "coastal vessels" to "the coastal vessels".

line 245, change "suggests" to "suggest";

Reply: Revised as suggested, Line 279, replaced "suggests" to "suggest".

line 246, change "SOAFPs of VOCs from ship exhausts" to "SOAFP of the VOCs from ship exhaust";

Reply: Revised as suggested, Line 281, replaced "SOAFPs of VOCs from ship exhausts" to "SOAFP of the VOCs from ship exhaust".

line 250, change "Like" to "Similar to";

Reply: Revised as suggested, Line 286, replaced "Like" to "Similar to".

line 251, change "SOAFPs" to "SOAFP";

Reply: Revised as suggested, Line 287, replaced "SOAFPs" to "SOAFP".

line 252, change "to interpret" to "in interpreting";

Reply: Revised as suggested, Line 288, replaced "to interpret" to "in interpreting".

line 252, delete "(IVOCs)";

Reply: Revised as suggested, Line 288, deleted "(IVOCs)".

line 261, change "higher" to "higher than that of the other ships,";

Reply: Revised as suggested, Line 298, replaced "higher" to "higher than that of the other ships".

line 262, change "has: to "had";

Reply: Revised as suggested, Line 299, replaced "has" to "had".

line 262, change "river vessels" to "the river vessels";

Reply: Revised as suggested, Line 299, replaced "river vessels" to "the river vessels".

line 262, change "test ships" to "the tested ships";

Reply: Revised as suggested, Line 300, replaced "test ships" to "the tested ships".

line 262, change the value" to "a value";

Reply: Revised as suggested, Line 300, replaced "the value" to "a value".

line 273, change "one the three ECAs newly established" to "one of the three newly established

ECAs";

Reply: Revised as suggested, Line 315, replaced "one the three ECAs newly established" to "one of the three newly established ECAs".

line 280, change "of EFs" to "in the EFs";

Reply: Revised as suggested, Line 322, replaced "of EFs" to "in the EFs".

line 280, change "more" to "the more";

Reply: Revised as suggested, Line 323, replaced "more" to "the more".

line 288, change "did bring about largely" to "resulted in substantially";

Reply: Revised as suggested, Line 332, replaced "did bring about largely" to "resulted in substantially".

line 289, change "for" to "due to";

Reply: Revised as suggested, Line 334, replaced "for" to "due to".

line 290, change "coastal vessels" to "the coastal vessels";

Reply: Revised as suggested, Line 334, replaced "coastal vessels" to "the coastal vessels".

line 290, change "is" to "was";

Reply: Revised as suggested, Line 334, replaced "is" to "was".

line 290, change "use" to "had used";

Reply: Revised as suggested, Line 335, replaced "use" to "had used".

line 290, change "all along and thus" to "the entire time and thus were";

Reply: Revised as suggested, Line 335, replaced "all along and thus" to "the entire time and thus were".

line 291, change "much higher" to "high";

Reply: Revised as suggested, Line 336, replaced "much higher" to "high".

line 291, change "larger emission" to "high level of emissions";

Reply: Revised as suggested, Line 336, replaced "larger emission" to "high level of emissions".

line 292, delete "would";

Reply: Revised as suggested, Line 337, deleted "would".

line 292, change "how to further lower the emission" to "and further lowering the emissions";

Reply: Revised as suggested, Line 337, replaced "how to further lower the emission" to "and further lowering the emissions".